# Nonhomologous end-joining uses distinct mechanisms to repair each strand of a double strand break

Adam J. Luthman[1,3], Kishore K. Chiruvella[1,3], Andrea M. Kaminski [2], Thomas A. Kunkel [2], Lars C. Pedersen [2] & Dale A. Ramsden [1]✉

Nonhomologous end-joining repairs chromosomal double strand breaks, but it is unknown whether both strands are repaired by this pathway, and if one strand break's repair path impacts the other. Here, we show that nonhomologous end-joining employs both of two a priori possible strategies. Strand breaks that can be directly ligated are joined near-simultaneously, with no effect of one strand break's repair path on the other. More complex end structures require obligatorily ordered repair. The first strand to be repaired is used as template for repair of the opposite/second strand break, with the latter repair reaction occurring fastest when also coupled to nonhomologous end-joining. Enforced asymmetry in repair of each strand break can extend to the gap-filling polymerase employed, and whether the polymerases incorporate RNA or DNA. Our results resolve questions about pathway mechanism and identify a requirement for flexibility of the nonhomologous end-joining machinery for efficient repair of both strand breaks within diverse cellular double strand breaks.

In mammals, nonhomologous end joining (NHEJ) is the primary pathway for repair of double-strand breaks (DSBs) generated by exogenous agents like ionizing radiation. It is also the only significant pathway for repair of DSB intermediates in V(D)J recombination, a process that assembles antigen-specific receptors in immune cells. These biologically relevant sources of DSBs have diverse end structures, varying from ends that can be directly ligated to more complex end structures that must first be processed by a variety of factors that include polymerases, end-cleaning enzymes, and nucleases (reviewed in ref. 1).

DNA synthesis activity – both nucleotide addition and ligation – associated with NHEJ must therefore be sustainable opposite a strand break, and typically mispairs, gaps, or damage as well. The ability to sustain synthesis activity in this context relies in part on diverse protein complexes of NHEJ core factors that bridge the two ends together, effectively acting as a surrogate for an intact template strand (Fig. 1a, inset)[2–6]. These core factors include the heterodimer of Ku70 and

Ku80, DNA-PKcs, LIGASE4 (Lig4), XRCC4, and XLF (XRCC4-like factor). The complexes formed by these core factors can additionally recruit factors for end processing, including Pol μ and Pol λ (two widely expressed members of the Pol X family)[7–9]. These complexes elegantly address the defining problem of NHEJ by bridging two DSB ends with near-perfect two-fold symmetry[10–15] (reviewed in refs. 16,17).

Left unaddressed are questions unique to this pathway and central to the understanding of pathway mechanism, and especially what role the end-bridging complexes play in directing the steps to complete repair. Does the two-fold symmetry of many end-bridging complex structures extend to their repair activity, such that each of the two strand breaks is repaired in parallel (effectively co-incident) and independent of the other (Fig. 1a, model 1: parallel, symmetric)? Alternatively, repair of the two-strand breaks could be obligatorily ordered, with the 1st strand to be repaired, then employed as a template strand for repair of the 2nd using conventional semi-conservative

[1]Department of Biochemistry and Biophysics, Lineberger Comprehensive Cancer Center, University of North Carolina at Chapel Hill, Chapel Hill, NC, USA. [2]Genome Integrity and Structural Biology Laboratory, National Institutes of Environmental Health Sciences, 111 TW Alexander Drive, Durham, NC, USA. [3]These authors contributed equally: Adam J. Luthman, Kishore K. Chiruvella. ✉e-mail: dale_ramsden@med.unc.edu

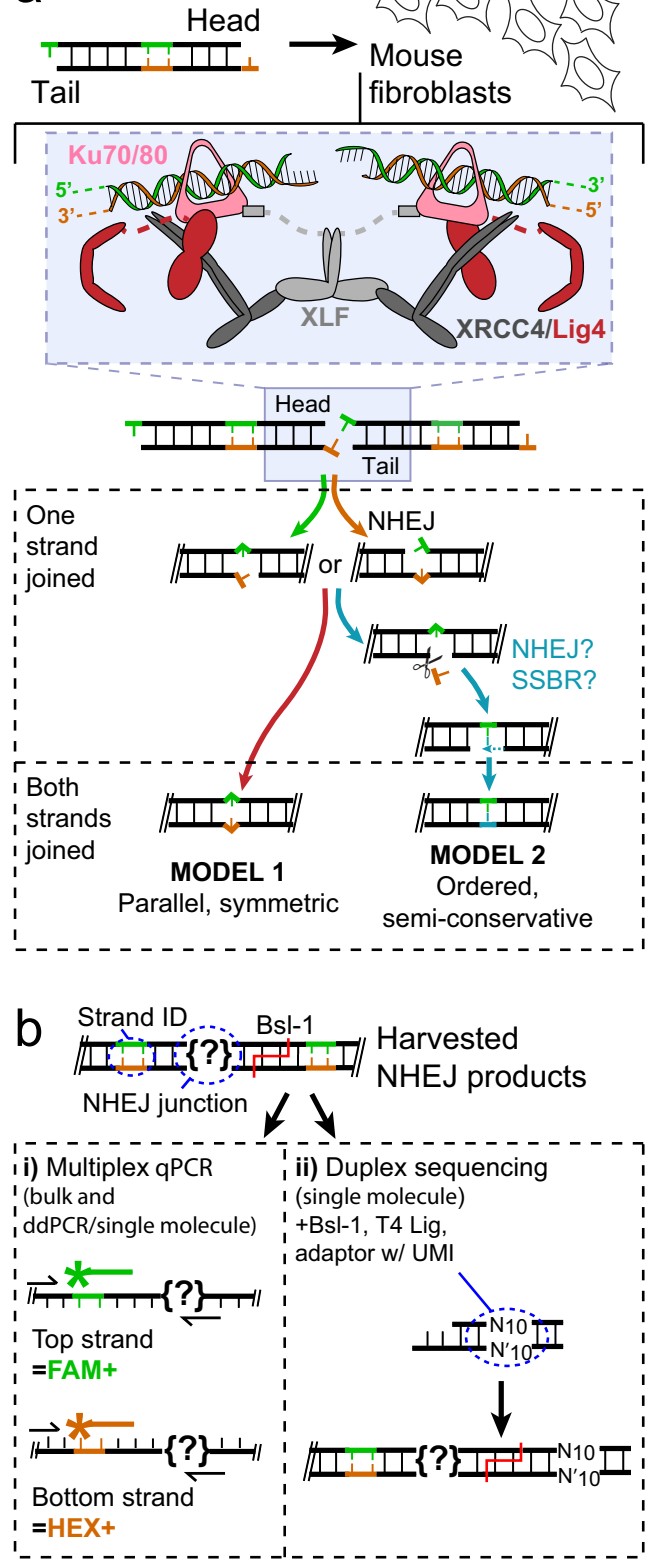

**Fig. 1 | Strand-specific repair of a double-strand break by cellular NHEJ. a** Linear substrates were introduced into cells. Substrate head and tail ends have mispairs and damage in the top (green) or bottom (orange) strand after alignment by a complex of NHEJ core factors (blue shaded inset; cartoon derived from structure in ref. 12), and can be repaired by two different paths that generate distinct products. **b** In NHEJ products (?) the top strand (green) was distinguished from the bottom strand (orange) by sequence differences in a mismatched region embedded in the substrate (strand ID), either i) during qPCR, using strand-specific probes with different fluorescent labels (Top strand, FAM+ probe; bottom, HEX + ) or ii) during analysis of sequences of NHEJ products, after ligation of a double stranded sequencing adapter that contained a unique molecular identifier (UMI) consisting of 10 random nucleotides ($N_{10}$).

use methods that allow identification of both which strand is being repaired, as well as the sequence of that strand's repair product (Fig. 1b), with single-molecule resolution. We show NHEJ surprisingly employs both of the two a priori possible strategies we described above, and that the strategy used is determined by the structure of the ends to be repaired. We further show that for ordered, semi-conservative repair (model 2), the NHEJ core factor complex continues to have an important role in guiding repair of the 2nd strand break, even though 2nd strand break repair could in principle be mediated by, e.g., SSBR.

## Results

### Parallel, independent repair of the two-strand breaks within a single DSB

To address how each strand of a DSB is repaired by cellular non-homologous end-joining (NHEJ), we needed to assess both the identity of the strand being repaired as well as the sequence of that strand's repair product. We therefore introduced linear NHEJ substrates (Supplementary Dataset 1) into transformed mouse embryo fibroblasts (MEFs), under conditions where head-to-tail joining of substrate ends saturates at physiologically relevant levels (tens of repair products/cell) in a physiologically relevant amount of time (tens of minutes) (e.g., Fig. 2b). Repair measured this way is largely specific to NHEJ as only trace levels of repair were detected in cells defective in any of the NHEJ core factors (Supplementary Fig. 1a–c and Supplementary Table 1), and repair was not affected by deficiency in the other major end joining pathway (Supplementary Fig. 1d). An embedded 5 nucleotide mispaired region (Fig. 1b, strand ID), was further used to provide annealing sites for two different fluorescent PCR probes, each specific to one of the two strands. The top strand-specific probe was a different color (fluorescein; FAM + ) than the bottom strand-specific probe (hexachlorofluorescein; HEX + ), allowing for strand-specific quantification of end-joining in the same multiplex quantitative PCR (Supplementary Fig. 1e). We confirmed the strand-identifying mispaired region was stable for the duration of the assay (Supplementary Fig. 1f) and did not interfere with NHEJ (Supplementary Fig. 1g).

In accord with past work[18], cellular NHEJ of ends possessing symmetric 5′ overhangs with terminal G:T mispairs (Fig. 2a) primarily generated repair products by direct ligation (i.e., bypassing the mispair that could interfere with ligation; 96% of all top strand junction sequences in this experiment, Supplementary Table 2). We focus here on whether a fully repaired duplex with this product is generated following model 1 vs. model 2 paths (Fig. 2a), by separately tracking the accumulation of different top strand (FAM+ products) as well as bottom strand (HEX+ products) with increasing time in cells. Product-specific primers (Supplementary Fig. 2a), together with strand-specific probes (Supplementary Fig. 1e), allowed us to quantify all four possible products associated with direct ligation of both strands. Strikingly, direct ligation products accumulated rapidly and efficiently (saturated at 10 min) on each strand (Fig. 2B, green and orange lines). By comparison, there were only trace levels of products consistent with semi-

DNA synthesis (Fig. 1a, model 2: ordered, semi-conservative). For this latter strategy, repair of the 2nd strand could be independent of NHEJ and instead rely on single-strand break repair (SSBR).

Here, we investigate the relationship between repair of each of the two strand breaks of a DSB during repair by NHEJ, as well as the importance of the end-bridging complex in directing each strand's repair. We track the repair of each strand in living cells over time. We

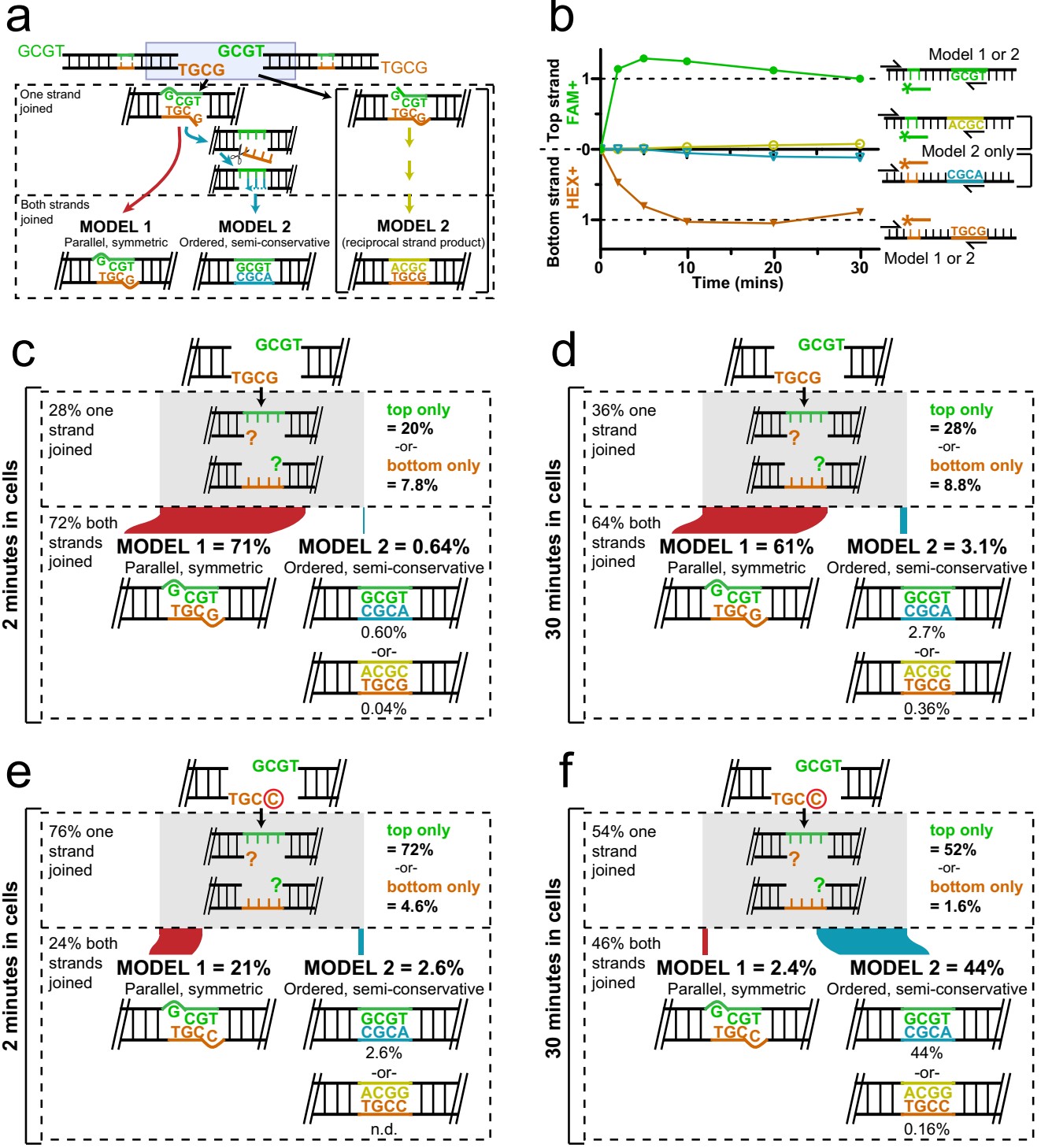

**Fig. 2 | Strand-specific repair of ends joined by direct ligation. a** Repair by NHEJ of the described substrate by model 1 vs. model 2 paths generates different products with both strands joined. **b** The substrate in panel a was introduced into T-antigen-transformed mouse embryo fibroblasts (MEFs), DNA recovered after increasing time in cells (minutes), and the amounts of repair products were measured by qPCR. Greater repair is represented by increasing upwards from the y-axis origin for the top strand or increasing downwards from the origin for the bottom strand. The amounts of directly ligated strands (filled symbols) are plotted for the top strand in green and the bottom strand in orange. Products using an opposite strand as a template (model 2, open symbols) are plotted for the top strand in yellow and the bottom strand in blue. All repair amounts are scaled relative to the most abundant product (top strand direct) at 30 min as determined by digital droplet PCR. **c, d** Repair products for the substrate used in panel b after 2 min (**c**) or

30 min (**d**) in cells were characterized by duplex sequencing. The frequencies of products with one strand joined (single strand break intermediates) and each of the three repair products with both strands joined are noted and represent the mean of $n = 3$ independent biological replicates. The width of line segments connecting one-strand joined intermediates to the different products with both strands joined reflects the proportion of molecules repaired by the model 1 path in red, or the sum of both model 2 path products in teal. **e, f** As in panel c, except using a substrate where a C was substituted for G at the 5′ terminus of only one of the two ends (circled), and comparing repair products recovered from cells after 2 min (**e**) or 30 min (**f**) in cells. The frequencies of products represent the mean of $n = 3$ independent biological replicates as defined for panel c. N.d. = not detected. Source data for all relevant panels are provided as a Source Data file.

conservative synthesis using either of these two direct ligation products as a template (model 2 products, Fig. 1a), even when NHEJ was assessed after repair was saturated (Fig. 2b, yellow and blue lines).

These bulk qPCR results are consistent with parallel and symmetric ligation of both strand breaks (model 1) in almost all products. However, they could be equally explained if only the top or the bottom strand of a given duplex molecule was joined (one strand joined, Fig. 1a). We therefore further analyzed repair products by first ligating an adapter that included a 10 random nucleotide unique molecule identifier (UMI, Supplementary Table 3)[19,20], then performing next generation sequencing on amplified products (Fig. 1b, Supplementary Tables 4-6). We ensured only thousands of input duplex molecules were present in each experiment. With more than a million ($4^{10}$) possible UMIs, we can thus infer with high confidence that top and bottom strand sequences with a shared UMI originated from the same duplex product molecule. Similarly, the absence of an opposite-strand product with the same UMI can be inferred as a molecule with only one strand repaired.

We applied this methodology to a pool of NHEJ products generated before and after saturation of cellular repair (2 and 30 min). Repair products associated with double-stranded substrate primarily had both strands joined after only 2 min in cells (Fig. 2c, 72%, compared to 28% one-strand joined intermediates). Moreover, in accord with bulk qPCR results (Fig. 2b), very low levels of semi-conservative (model 2) products were observed even after extended incubation in cells (3.1% after 30 min, Fig. 2d). We conclude that for this substrate there is a dominant role for parallel and symmetric repair of both strands (model 1 favored >20:1 over model 2, Fig. 2c, d).

## Ordered, semi-conservative repair of the two-strand breaks within a single DSB

We reasoned that the symmetric obstacles to ligation in this previous experiment may have favored a symmetric repair path. We therefore substituted a 5′ terminal C (circled, Fig. 2e, f) for G in the bottom strand only. After end alignment, this substrate retains a G:T mispair at the top strand break terminus (as in Fig. 2a-d), but now has a C:T mispair at the bottom strand break terminus. The primary repair product for the top strand continues to favor direct ligation of the terminal G:T mispair (96% of all top strand repair products, Supplementary Table 7). In contrast to Fig. 2b, c results, however, the majority of bottom strand molecules remain unrepaired after 2 min in cells (72%, compared to 20% for substrate used in Fig. 2c). We confirm this reflects less efficient direct ligation of C:T mispairs, relative to G:T mispairs, as we similarly observed inefficient direct ligation of a C:T mispair whether it was located in the bottom strand (Fig. 2e, f), the top strand, or both (Supplementary Fig. 2b-e). More importantly, the bottom strand was efficiently repaired only after extended time in cells (30 min; 46.5%), and most of the resulting products for this strand were now consistent with semi-conservative DNA synthesis, using the directly ligated top strand as template (Fig. 2f; 44%). We conclude that a single nucleotide substitution was sufficient to switch the repair mechanism, relative to the substate with symmetric G:T mispairs: the ordered, semi-conservative path to repair is now dominant (model 2 favored 18:1 over model 1).

We considered next an end structure that requires end processing before ligation, employing a substrate with symmetric non-complementary 3′ overhangs (GAG-3′). In accord with past work[21], this class of end structure is frequently repaired by ligation after Pol μ-mediated addition of a single nucleotide (Fig. 3a, left substrate) (2nd most common product, Supplementary Table S8, Supplementary Fig. 3a). For this end structure a C is added to one end (+C product, Fig. 3a), as it is complementary to the nucleotide upstream of the other end's 5′ terminus and sufficient to promote ligation. We confirmed that this product is Pol μ-dependent (Fig. 3b, left, and Supplementary Fig. 3b−d). In accord with past work, Pol μ-dependent products with

longer complementary sequence additions were comparatively rare (+ TC, +CTC) (Supplementary Fig. 3a), and non-complementary additions were not detected (+ T, +A, or +G; $<1 \times 10^{-4}$).

Prior work indicates two members of the Pol X family, Pol μ and Pol λ, interact with NHEJ core factors within the end-bridging complex through their N-terminal BRCT domains[7,8]. We addressed the extent to which Pol μ activity is dependent on interaction with these core factors by introducing a mutation in a single hydrophobic residue on the surface of the BRCT domain−Pol μ F46A−that disrupts its recruitment to the NHEJ complex, but has no impact on its intrinsic catalytic activity[7,8]. Structures of end-bridging complexes indicate Pol μ F46A (or the structurally analogous Pol λ L60A) will disrupt a hydrophobic interaction with Ku70 F303[15,22](modeled in Fig. 3c). We show here that introduction of wild-type Pol μ, but not Pol μ F46A, into $Polm^{-/-}$ cells promotes generation of the +C NHEJ product (Fig. 3b, right). We conclude that catalytically active Pol μ alone isn't sufficient for this repair reaction−Pol μ must also interact with Ku in the end-bridging complex to contribute to NHEJ.

We then addressed how NHEJ proceeds at the strand-specific level for this substrate (Fig. 3a). We show the Pol μ-dependent +C product accumulates rapidly on the top strand (Fig. 3d, top strand; 91% after 5 min in cells). Bottom strand molecules that are complementary to the top strand +C product (Fig. 3d, bottom strand; model 2 products) initially accumulate more slowly (4% at 5 min), but by 30 min have accumulated to levels nearly equivalent with top strand product (86%; Fig. 3d, Supplementary Fig. 3e). We confirmed the reciprocal bottom strand +C product also efficiently follows a model 2 repair path, in which the top strand uses the repaired bottom strand as template (82%; Supplementary Fig. 3e). Our results are indicative of a dominant role for obligatorily ordered, semi-conservative repair of the 2nd strand, using the intact 1st strand as template (model 2).

Pol μ most often adds ribonucleotides (RNA) during cellular NHEJ, and addition of RNA instead of DNA is essential for accurate and efficient joining by NHEJ of this class of substrates[23] (also Supplementary Fig. 3f). Consistent with these prior observations nearly all top strand +C products were initially sensitive to alkali treatment (Fig. 3D, purple fraction in top strand repair; 95% alkali-sensitive after 5 min), which cleaves nucleic acid chains with embedded ribonucleotides (Supplementary Fig. 3g). The fraction of RNA-embedded top strand products was reduced to 61% after 30 min in cells, but not when cells were deficient in RNAseH2 (Supplementary Fig. 3h; 91% after 30 min in cells), the enzyme essential for removing RNA embedded in genomic DNA[24,25]. By comparison, accumulated bottom strand semi-conservative products had negligible alkali sensitivity throughout (Fig. 3D, purple fraction in bottom strand repair), indicating they were purely DNA. We conclude there are three successive strand break repair reactions: (i) addition of a ribonucleotide C and ligation of the top strand (1st strand repair), then (ii) repair of the bottom strand with purely DNA using the RNA-embedded 1st strand as template (2nd strand repair), then (iii) replacement of the RNA embedded in the 1st strand with DNA by RNAseH2-dependent ribonucleotide excision repair (RER).

## Role of NHEJ in 2nd strand repair

We sought to resolve whether the NHEJ core factor complex plays a role in how the 2nd strand is repaired, given that it is no longer a DSB. We employed a substrate related to that used in Fig. 3b, d (compare left/gray and right halves of Fig. 4a), except i) the Pol μ requirement for 1st strand repair was suppressed by pre-adding the ribonucleotide C (Supplementary Fig. 4a), and ii) 2nd strand repair now requires only gap-filling (+CTC) and ligation (Supplementary Fig. 4b). These modifications allowed us to focus on the effectiveness of each gap-filling polymerase in 2nd strand repair by using cells deficient in all three Pol X family members (Pol μ, Pol λ, and Pol β) (Supplementary Fig. 4c, d), then comparing results to experiments where each Pol X member was

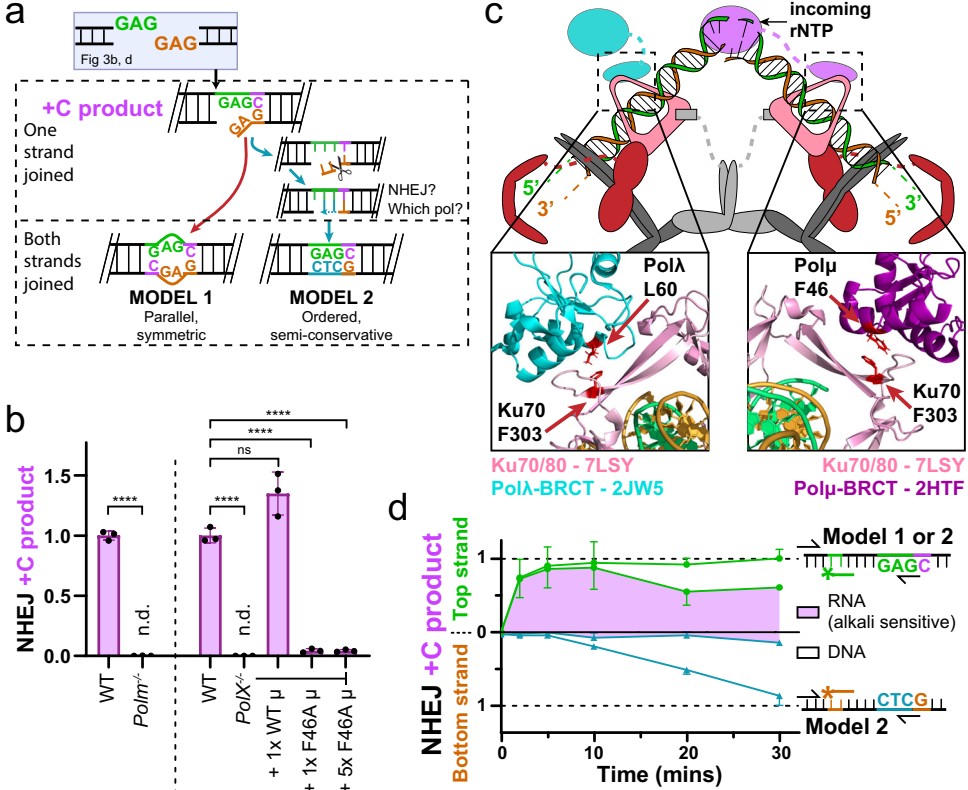

**Fig. 3 | Strand-specific repair of ends requiring processing before ligation. a.** Repair by NHEJ of substrates with non-complementary 3′ overhangs generates different products if repaired by model 1 or model 2 paths to both strands joined. **b** The substrate in panel a (3′-GAG/GAG) was introduced into wild-type MEFs (WT), MEFs deficient in only Pol μ (*Polm⁻/⁻*), MEFs deficient in Pol μ, Pol λ, and Pol β (*PolX⁻/⁻*), and *PolX⁻/⁻* MEFs complemented by the introduction of wild-type Pol μ (1xWT μ, 1x = 100 ng), the same amount of Pol μ F46A (1xF46A μ), or five times the amount of Pol μ F46A (5xF46A μ). Vertical dashed line separates independent experiments. Data are the mean amounts of +C NHEJ product, as shown in panel a, ±s.d. from $n = 3$ independent biological replicates, relative to WT cells. Means were analyzed by unpaired t-test (left of dashed line) or one-way ANOVA with Dunnett's test for multiple comparisons (right of dashed line). ****$p < 0.0001$, ns = not significant, n.d. = not detected. **c** Alphafold models[42] of the Pol λ (cyan, left inset; PDB ID code

2JW5)[8] or Pol μ (magenta, right inset; PDB ID code 2HTF)[7] interaction with Ku (rose; based on PDB ID code 7LSY)[12], with the hydrophobic residues required for this interaction in red stick representation. **d** The substrate in panel a was introduced into MEFs. Relative strand-specific repair plotted as in Fig. 2b except using the noted substrate and plotting top strand Pol μ-dependent repair ( + C product, defined in panel a, green), bottom strand repair using this top strand product as template (blue), and the fraction of both products sensitive to alkali treatment (shaded purple) compared to total (mock-treated) product. Data are the mean ± s.d. of $n = 3$ independent biological replicates, with the exception of 30 min top strand fraction RNA containing 2 replicates, with all product amounts expressed relative to the total (mock treated, purely DNA and RNA-embedded products both) top strand Pol μ-dependent product after 30 min in cells, as determined by digital PCR.

added back one at a time. Some 2nd strand repair is notably observed even in the absence of all three X family polymerases (19% at 5 min, Fig. 4b; 49% at 30 min, Supplementary Fig. 4e), suggesting that non-X family polymerases, though less effective, have some activity at this step. Importantly, rapid 2nd strand repair was fully recovered upon introduction of Pol λ in Pol X deficient cells, while equivalent amounts of either Pol μ or Pol β had no effect (Fig. 4b). The Pol λ L60A variant specifically defective in interaction with NHEJ core factors (Fig. 3c, cyan)[8,22] was also relatively inactive in 2nd strand repair, even when introduced in excess (Fig. 4b, Supplementary Fig. 4e). This is consistent with an important role for the NHEJ core factor end-bridging complex in directing repair of the 2nd strand.

The substrate context addressed here requires Pol λ to retain activity on a substrate where a ribonucleotide is embedded in the template strand, opposite the 3′ primer terminus. To address how this is accommodated, we solved a crystal structure of Pol λ on such a substrate (Fig. 4c–f; ribonucleotide in magenta, PDB ID code: 9NPU), and compared it to a prior structure using a purely DNA template[26] (Fig. 4e, PDB ID code: 7M07, template with embedded ribonucleotide colored, compared to purely DNA template in gray). There is little impact of the embedded ribonucleotide on the location of either the ribonucleotide base (Fig. 4e, position −1) or the adjacent templating

nucleotide (Fig. 4e, position 0). Accordingly, both the primer terminus (including the attacking 3′OH nucleophile) as well as the incoming nucleotide are positioned consistent with a catalytically active complex (Fig. 4f).

## Discussion

Implicit in the repair of double-strand breaks is the need to repair both. We show here how this is accomplished during cellular non-homologous end joining. This pathway's emphasis on complete repair of the DSB is biologically significant, as a half-repaired DSB will contribute to genome instability if encountered by a replication fork[27], and is indeed the starting substrate for many other repair pathways. As introduced above, there are a priori only two possible strategies (models) to complete repair. We show that cellular NHEJ surprisingly uses both, depending on substrate context, with important implications for our understanding of the pathway mechanism. We discuss below the possible factors that determine the strategy chosen, as well as the role of end-bridging complexes in guiding these disparate paths to repair.

Ends with symmetric, terminal G:T mispairs are directly ligated by NHEJ without prior processing of the ends[18,28]. We show that this occurs for both strands of a single DSB in parallel (i.e., almost simultaneously

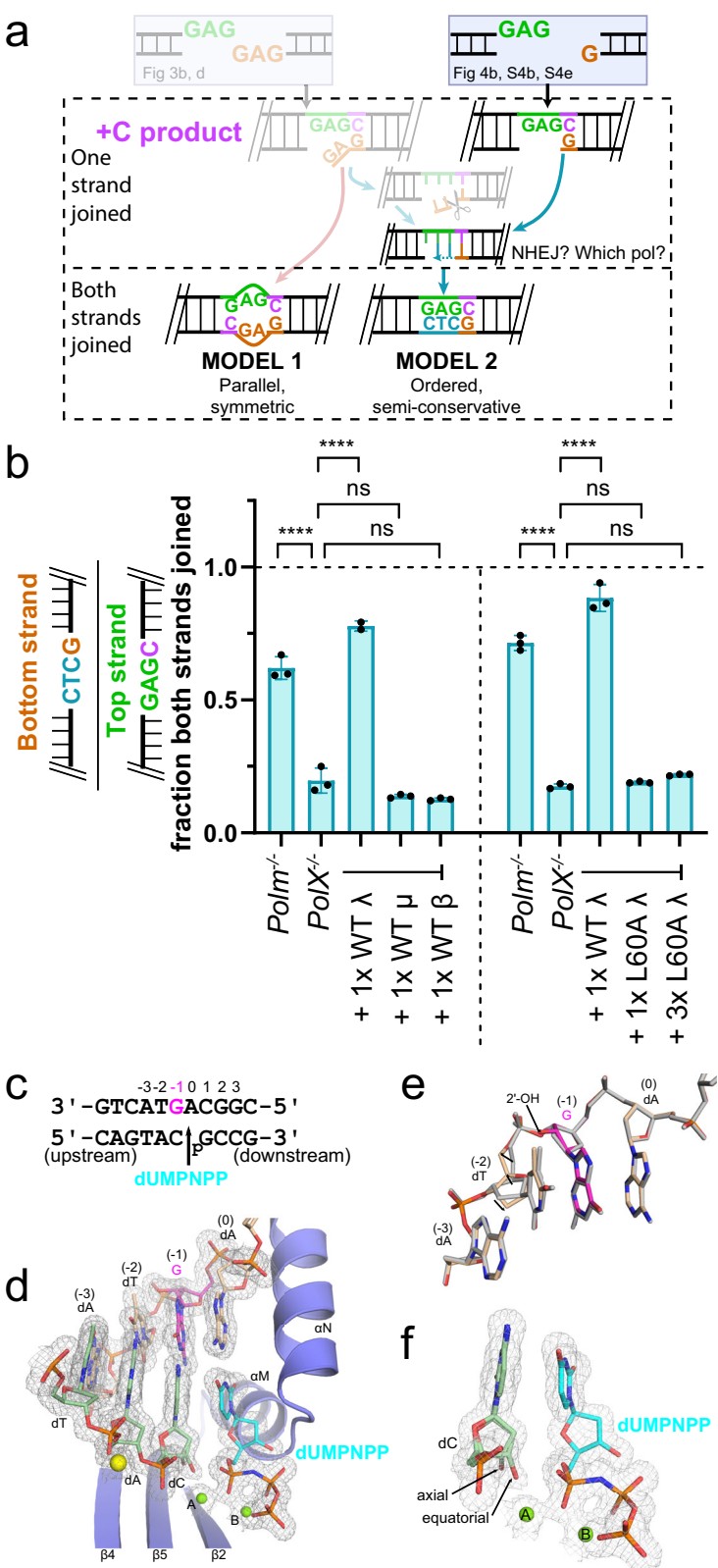

(Fig. 1a, model 1: parallel, symmetric). This mechanism is likely also employed for cellular repair of DSBs with blunt ends or ends with fully complementary overhangs, consistent with in vitro studies[12]. Moreover, past work argues that similar parallel direct ligation will occur when there are similarly subtle and symmetric barriers to ligation. This includes symmetric terminal 8-oxo-7,8-dihydroguanine (8oxoG)[18,28], the most abundant radiation-associated base damage, and possibly even Topoisomerase II adducted breaks. The mechanism described here can notably leave mispairs or base damage in the joined product. The different products on each strand must then be resolved by base excision repair or mismatch repair. Alternatively, they could interfere with replication or transcription.

More complex end structures were primarily repaired in obligatorily ordered steps (model 2), where one strand break was repaired

**Fig. 4 | Second-strand repair by NHEJ. a** Substrate employed to focus on repair of gap in 2nd strand of substrate in Fig. 3. **b** The substrate on right from panel a was introduced into MEFs deficient in Pol μ only (*Polm*[-/-]), or in all three members of the Pol X family (*PolX*[-/-]), or *PolX*[-/-] deficient cells complemented by introduction of equal amounts of Pol β (1xWT β, 1x = 250 ng), Pol μ (1xWT μ), Pol λ (1xWT λ), Pol λ L60A (1xL60A λ), or 3-fold more Pol λ L60A (3xL60A λ). A vertical dashed line separates experiments that were performed on different days. DNA was harvested after 5 min in cells. The abundance of top strand products with the noted +C product, as well as the abundance of bottom strands with the complementary sequence, was determined by digital PCR. The fraction of both strands joined was expressed as the ratio of bottom strands with complementary sequence over top strand +C products, ±s.d. from n = 3 independent biological replicates, with the exception of *PolX*-/- +1x WT Pol λ (left of dashed line), including two replicates. Means were compared by one-way ANOVA with Dunnett's test for multiple comparisons for each experiment independently (each side of the dashed line). ****$p < 0.0001$, ns = not significant. Source data for all relevant panels are provided

as a Source Data file. **c** 1nt-gapped SSB DNA intermediate substrate with template strand embedded ribonucleotide (magenta) co-crystallized with the Pol λ catalytic domain and an incoming nonhydrolyzable dUMPNPP nucleotide (cyan). **d** View of the pre-catalytic SSB active site (protein in blue, template DNA strand in khaki, upstream primer strand in green, incoming nucleotide in cyan) with secondary structures depicted as ribbons, and bound substrates in stick. The Na[+] ion bound in the HhH2, and the active site divalent Mg[2+] ions are shown as yellow or green spheres, respectively. $2F_o$-$F_c$ electron density is drawn in gray mesh (contoured at 1σ). **e** Superposition of the 2nd-strand gap-filling intermediate (khaki) containing template-embedded ribonucleotide (magenta) with the same substate but containing a deoxyribonucleotide at the same position (gray, PDB ID code: 7M07 [https://doi.org/10.1038/s41467-022-31278-4])[26], highlighting slight structural variations in the template strand due to the presence of the ribonucleotide. Positional shifts and alterations in sugar puckering are marked by black arrows. **f** Zoomed-in view of the upstream primer terminal (light green) and incoming dUMPNPP (cyan) nucleotides, highlighting the mixture of conformations for the 3'-OH.

first, then used as a template for repair of the opposite/2nd strand break. For substrates requiring initial end processing by Pol μ, switching from 1st to 2nd strand repair was associated with switching the polymerase used: Pol μ for 1st strand repair (Fig. 3a, b)[21], then Pol λ for 2nd strand repair (Fig. 4b). Pol μ and Pol λ are similarly recruited by the NHEJ core factor complex (Figs. 3b, c and 4b). We conclude that either there is a mechanism favoring recruitment of both polymerases (a different one for each side of an end-bridging complex) or that the polymerase(s) recruited are readily exchanged. The switch in the polymerase used is dictated by substrate differences. Synthesis from an unpaired primer, required for 1st strand repair of the substrates in Figs. 3a and 4a, is dependent on Pol μ (Fig. 3b and Supplementary Fig. 4b) (in accord with past work[21]). By comparison, Pol μ is several orders of magnitude less active than Pol λ on gapped substrates with paired primers[29,30], as required for 2nd strand break repair. A consequence of this polymerase switch is that while RNA is embedded during 1st strand repair, 2nd strand repair involves purely DNA synthesis (Fig. 3d). We suggest this is beneficial, since RNA embedded in both strands would risk re-breaking the chromosome during attempted replacement of ribonucleotides by ribonucleotide excision repair.

For all the end structures tested here, there was a disproportionate favoring of one path over the other. Model 1 was favored 20:1 in Fig. 2b-d experiments, and model 2 was favored >18:1 for Fig. 2e, Figs. 3d, 4b, and Supplementary Fig. 4e experiments. Thus, while both possible strategies can be used depending on end structure, the extreme favoring of one path over the other argues that there is little flexibility in the choice of path for a given end structure. Comparison of results in Fig. 2c, e suggests a symmetric strand break context may be important in directing repair towards a symmetric, model 1 path. However, symmetry in the strand break context alone isn't sufficient, since the symmetric end structure employed in Fig. 3 was also repaired by the asymmetric model 2 path. We suggest a need for end processing also helps dictate the path to repair. Whether end processing factors are engaged is determined primarily by attempted ligation (Lig4 catalytic domain binding to DNA end, Fig. 5 step 1)[4,18,28,31]. Repair of each strand break is likely organized the same way (attempted ligation first), as this is the most straightforward way to explain why the example where both strand breaks can be ligated without processing favors a model 1 path to repair (Fig. 2a-d).

The model 1 path employed for Fig. 2a-d experiments requires successive Lig4-mediated ligations of each of the two strand breaks within minutes (Fig. 5). Additionally, existing cryo-EM structures all locate the catalytically active strand break at the same location relative to the rest of the complex, at the top, opposite the XLF/XRCC4/Ku protein interaction framework that bridges the two ends (Fig. 5, 2nd step of model 1)[12,15]. These two cited studies further note that successive ligations of the two strand breaks are best achieved if the DNA

helix moves (rotates or changes twist, and/or translocates), relative to the rest of the complex. This motion of the DNA is necessary if each strand break is to be ligated at the equivalent position within the complex, as the structural studies imply (Fig. 5, step 3 of model 1, see Supplementary Movie 1 for animated version). It has also been suggested that repair of the 2nd strand takes advantage of the two-fold symmetry of many short-range complexes, such that its ligation is mediated by the second of the two ligase molecules that were present in the initial complex[12,15]. However, single-molecule studies indicate that transition to the ligation-competent short-range complex is almost always preceded by reduction of the number of ligase molecules to one[31]. This latter result implies that the resolution of the 2nd strand break requires the rapid acquisition of a second ligase molecule.

The model 2 path apparent in results in Figs. 3−4 show there are successive engagements of polymerase, then ligase, for each of the two strands in sequence (Fig. 5, right path, see Supplementary Movie 2 for animated version). This obligatorily ordered path (model 2) unsurprisingly takes longer (tens of minutes) than the model 1 path (minutes). A structural study argues that for a given strand, both polymerase and ligase-dependent steps employ polymerase and ligase molecules that are present on the same arm of a short-range complex – an active arm (vs. the other supporting arm)[15]. The idea that there is a built-in preference for finishing all steps on one strand before starting the other suggests that all end structures requiring a processing step before ligation may favor a model 2 path.

As with the model 1 path, the switching to 2nd strand repair is most efficiently accommodated by rotation (and possibly translocation) of the DNA helix after ligation of the 1st strand, then a switch in the identity of the active vs. supporting complex arm (Fig. 5, step 4 of model 2 path). Activity of Pol X family members generally requires much greater kinking of upstream vs. downstream helix paths (near 80°)[32] (e.g., inset to Fig. 5), relative to Lig4 activity (35°), and both can be accommodated by the end-bridging complex[15]. The ability to accommodate the associated repeated flexing of DNA within the complex presumably relies on mobility of Ku on DNA, flexible XRCC4 and XLF stalks, and a network of functionally important DNA-protein and protein-protein interactions that are connected through disordered tethers[15,33,34].

DNA-PKcs is required for early steps in NHEJ[2,35], but is missing from the catalytically active short-range complexes that mediate end processing and ligation steps[4]. The most parsimonious solution is that both strands are repaired within the same short-range complex (as in Fig. 5), arguing that DNA-PKcs would be dispensable for the transition to repair of the 2nd strand break. However, it remains possible that this transition requires an alternate DNA-PKcs-containing long-range complex[36], especially if processing of 2nd strand break ends requires DNA-PKcs kinase activity.

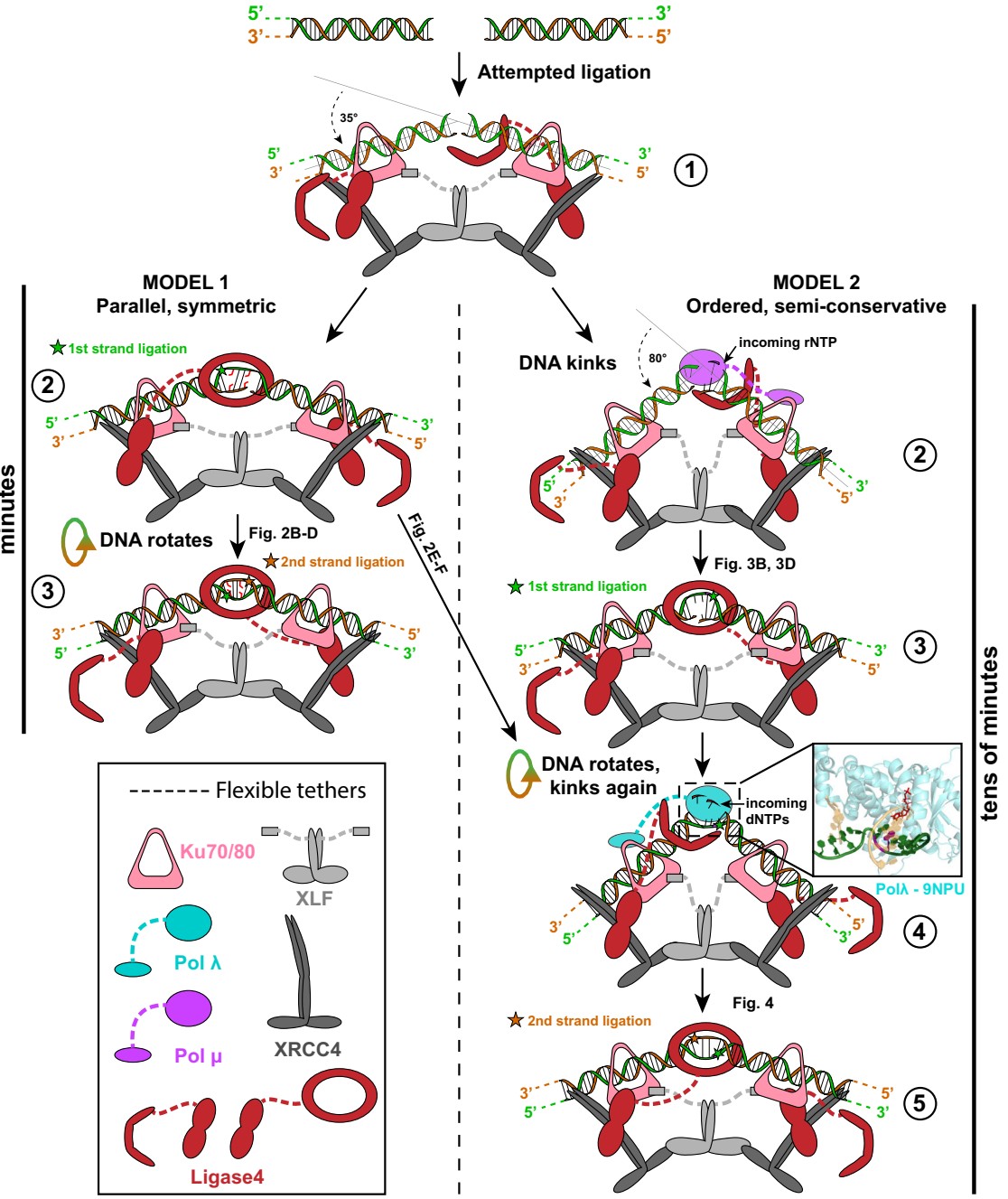

**Fig. 5 | Distinct mechanisms of repair by nonhomologous end-joining.** Model 1 path on the left, with parallel, symmetric ligations by Lig4 occurring in minutes with intervening DNA rotation. Model 2 path on the right, including Pol X-mediated

end processing, occurs in tens of minutes using sequential semi-conservative steps punctuated by DNA kinks and rotations during repair of each strand break.

In sum, our results emphasize how the NHEJ core factor complex adapts to differences in end structure even when considering the different strands of a single double-strand break, thus facilitating this pathway's ability to fully repair a diverse spectrum of DSBs.

## Methods

All materials are available upon request.

### Experimental rationale and limitations

We sought to analyze the mechanisms for repair of individual strands of a double-strand break by NHEJ over time in cells. We therefore employed extrachromosomal DSB substrates that could be electroporated into mouse fibroblasts (Fig. 1a), and ensured conditions were such that repair was appropriately dependent on relevant NHEJ factors

(Supplementary Fig. 1a-c) and complete on a physiologically relevant time scale (Figs. 2b, 3d, and Supplementary Figs. 3f, 4a) We further developed strand-specific (Supplementary Fig. 1E) and repair product-specific PCR assays (Supplementary Figs. 2a–d, 3b and Supplementary Dataset 1) and duplex sequencing, as outlined in Fig. 1b (see also Supplementary Tables 3-4). Combining these methodologies allowed detection of specific repair outcomes in each strand of a DSB in a time-resolved manner following NHEJ in living cells. This experimental model's ability to allow systematic variation of substrate end structure, initiation of cellular repair of these substrates with high synchrony (reactions start within minutes), and the ability to track each strand's repair product in parallel was essential. Indeed, the ability to systematically vary end structure will also be important for future work addressing other end structures. These substrates are

notably neither chromatinized nor fully restricted to nuclei upon introduction into cells, and we cannot exclude the possibility that this impacts the extent our observations can be generalized to chromosomal repair.

## Cell lines

Mouse embryonic fibroblasts (MEFs) and their variants (gift of Dr. L. Blanco) were derived from E14.5 d embryos, SV40 T-antigen transformed, and grown in Dulbecco's modified Eagle's medium (DMEM, Corning) containing 10% fetal bovine serum (VWR Life Sciences, Seradigm) and 100 µg/mL penicillin (Sigma) at 37 °C and 5% $CO_2$. *Polt$^{-/-}$Polm$^{-/-}$Polb$^{-/-}$* (*PolX$^{-/-}$*) MEFs were derived from previously described *Polt$^{-/-}$Polm$^{-/-}$* MEFs[21] by introduction of SpCas9-sgRNA targeting exon 7 of *Polb* (Supplementary Table 1 and Supplementary Fig. 4c, d). *Polt$^{-/-}$Polm$^{-/-}$* MEFs and *PolX$^{-/-}$* MEFs were grown in media additionally supplemented with 5 mM N-acetyl-L-cysteine (Sigma). *Ku70$^{-/-}$* MEFs were generated by introduction of Cas9-sgRNA targeting *Xrcc6* intron 4 and exon 6 (Supplementary Table 1) and were complemented by re-expression of Ku70 (+KU70) using lentiviral transduction (Supplementary Fig. 1b). *DNA-PKcs$^{-/-}$* MEFs were generated by introduction of Cas9-sgRNA targeting *Prkdc* exon 5 (Supplementary Table 1). HCT-116 cells and their variants[37–39] were the gift of Dr. E. Hendrickson (University of Virginia) and were grown in McCoy's 5 A medium (Corning) supplemented as above. *Polq*-deficient MEFs were the gift of Dr. R. Wood[40]. Spent media from cell cultures was routinely confirmed negative for mycoplasma contamination (less than 10 genomes/mL) using an rDNA amplicon.

## Western blotting

Whole-cell protein lysates were prepared using radio-immunoprecipitation assay buffer supplemented with protease inhibitors (Sigma, P8340). Thirty micrograms of protein lysate were loaded onto 4-12% Bis-tris polyacrylamide gels and resolved using MOPS-SDS running buffer. Proteins were transferred to a nitrocellulose blotting membrane in NuPAGE transfer buffer containing 10% methanol. Membranes were blocked in 3% fat-free milk for 2 h at room temperature, then incubated overnight at 4 °C in primary antibodies (Ku70 (1:1000) – Cell Signaling Technologies D10A7, Actin (1:5000) – Novus Biologicals nb600-535, Pol β (1:1000) – Abcam ab26343) diluted in 3% BSA. Membranes were washed with PBST and incubated with corresponding secondary antibodies (Licor) at a 1:10,000 dilution in 1% BSA for 2 h at room temperature. Probed membranes were imaged and analyzed on a Licor Odyssey machine.

## Extrachromosomal double-strand break substrate design

DNA double-strand break extrachromosomal substrates were obtained as single-strand oligonucleotides from Integrated DNA Technologies (ultramers, IDT) and annealed in 10 mM Tris, pH 7.5 supplemented with 100 mM NaCl and 0.1 mM EDTA. Substrate strands (sequences in Supplementary Dataset 1) contained 5 nucleotides of mismatched sequence, allowing differentiation of the two independent strands in each repaired substrate molecule, either by strand-specific fluorescent probes or by next-generation sequencing (see below).

## Single-strand break repair (SSBR) control design

A single-strand oligo was synthesized (IDT) containing two BtsI restriction sites in opposite orientations (sequence in Supplementary Dataset 1). This molecule was duplexed and amplified by 5 cycles of PCR using Phusion HF polymerase (New England Biolabs, NEB), then inserted into blunt II TOPO vector (Invitrogen) to generate a plasmid for electroporation efficiency control (pSSBR). pSSBR was then treated with 10 units of Nb.BtsI (NEB) at 37 °C for 4 h, generating a plasmid with single-strand breaks (nicks) on opposing strands, separated by a distance of 36 bp.

## Extrachromosomal substrate nucleofection

DSB substrates were electroporated at 40 nM (5'-GCGT/GCGT and 5'-GCGT/CCGT), 100 nM (3'-GAG/GAG and 3'-GAG/G), 1 nM (5'-GC/GC), or 0.025 ng of pSSBR. Nucleofections were performed with a protocol optimized for MEFs using the Neon electroporation system (Invitrogen) (cells resuspended in Gibco 1xPBS, 1350 mV, 1 pulse, 30 ms) or Amaxa nucleofection system (Lonza) with proprietary nucleofection protocols optimized for HCT116 or MEFs, and cells resuspended in Solution I (5 mM KCl, 15 mM $MgCl_2$, 50 mM Mannitol, 120 mM sodium phosphate solution pH 7.2). Complementation of cells with purified proteins (Pol µ, Pol λ, and Pol β and their variants) was carried out with 100–750 ng of protein added to the substrate transfection mixture immediately prior to electroporation. Cells were then incubated at 37 °C for the indicated time points in a mixture containing 1x Hank's balanced salt solution (HBSS, Gibco) or DMEM, each supplemented with 1 mM $MgCl_2$, and 125 U Benzonase nuclease (Sigma/Pierce). Reactions were stopped by extraction with a buffer containing sodium dodecyl sulfate (ATL, Qiagen), supplemented with 2 mg/mL Proteinase K (Qiagen), 40 µg RNaseA (Sigma), and 1 mM EDTA.

## Quantitative and digital PCR

Quantitative PCR was performed on purified genomic DNA using TaqMan Advanced Master Mix (Applied Biosystems), 250 nM primers and 333 nM probes (sequences listed in Supplementary Dataset 1), and a QuantStudio6 (Applied Biosystems). Threshold cycles (Ct) were calculated using QuantStudio real-time PCR software (Applied Biosystems, v1.7.2) and normalized to account for differences in electroporation and sample recovery using a spike-in control (e.g., 5'GC/GC or pSSBR). Statistical analyses compared normalized Ct data (Graphpad Prism 10) before transformation to linear scale representation with Excel (Microsoft).

Digital PCR was performed using droplet digital PCR (ddPCR, BioRad Laboratories) for quantification of bottom:top strand ratios (Figs. 2b, 3d, 4b, and Supplementary Figs. 2e, 3e, f, 4a, 4e) as well as for quantification of input template molecules for next-generation duplex sequencing (Fig. 2c–f). Droplets were generated for each sample using the Automated Droplet Generator (BioRad), amplified for 50 cycles with primer/probe annealing at 56 °C and extension for 1 min at 72 °C, then analyzed on the QX200 droplet reader for FAM and HEX channels. Data was analyzed with QX Manager 1.2 (BioRad).

Primers and probes for PCR applications were confirmed to specifically amplify individual strand products (Supplementary Figs. 1e, 2a, and 3b) and were confirmed to be linearly responsive to varying product amounts when used in multiplexed PCR reactions (Supplementary Fig. 3c, d).

## Single-molecule next-generation sequencing

A sequencing adapter containing 10 random nucleotides served as a unique molecular identifier (UMI). This length of UMI sequence generates 410 possible sequences, a roughly 200-fold excess over input substrate molecules, thus limiting the possibility of two molecules being appended with identical UMIs. The adapter consisted of a long strand containing the UMI and a shorter, annealed strand (IDT) to initiate linear extension across the random UMI sequence (Supplementary Table 3). Linear extension of 1 µM adapter was performed with 10 Units Klenow exo (NEB) for 5 min at 37 °C and was validated for successful duplexing by digestion with HphI (NEB).

DNA samples harvested from electroporations were digested with BslI and ClaI (NEB) to prepare ends for ligation to the sequencing adapter, then mixed with 20 pM duplexed adapter and 3000 Units of T7 DNA ligase (NEB). Linker-ligated DNA for each sample was then purified (QiaQuick PCR purification, Qiagen) and amplified for 28 cycles with primers that contained unique 6 nt barcodes, 0–5 nt of phasing, and Illumina P5/i5 or P7/i7 sequences (Supplementary Table 3). Following another purification step, library concentrations

were quantified, and samples were pooled in balanced proportions. The pooled library was then enriched with an additional 6 PCR cycles using P5 and P7 primers and gel-purified from a 1% agarose gel (Seakem) using the QiaQuick gel extraction kit (Qiagen). Finally, the pooled sample was cleaned with AMPure XP beads (Beckman Coulter), then sequenced (Azenta Life Sciences) using 2× 150-cycle paired-end reads.

### Next-generation sequencing analysis
Recovered sequences were trimmed, merged, and de-multiplexed using CLC Genomics Workbench version 24 (Qiagen). A custom Python script (PyCharm Community Edition 2021, JetBrains) was generated to annotate each read according to UMI, strand-specific identity, and repair junction sequence (Fig. 2c–f). The script was run on the Longleaf computer cluster (University of North Carolina at Chapel Hill). For each indexed library, we excluded UMIs where the number of reads for that UMI was less than 2% of the mean oversampling ratio (total reads/number of input template molecules), then further excluded strands for a given UMI if the read count was <20% of the sum of reads for that UMI. The resulting UMIs were then considered as a single dsDNA adapter-ligated molecule for subsequent analysis. The custom sequence analysis code is available at https://github.com/aluthman/Ramsden-Lab.git and at the permanent repository Zenodo (https://doi.org/10.5281/zenodo.15528811).

### Cloning, expression, and purification of full-length X family polymerases
Wild-type and mutant X-family polymerases were purified as in ref. 7 for Pol μ (WT and F46A)[8], for Pol λ (WT and L60A). Briefly, Pol μ variants were purified from *E. coli* via fractionation over SP-fast flow Sepharose (Pharmacia) and eluted into protease inhibitor AEBSF (Roche). Pol λ variants were cloned into the pGEX4T3 vector (GE Healthcare) containing a TEV protease site. Expressed proteins were purified from *E. coli* BL21 Codon-Plus (DE3)-RP over glutathione sepharose 4B resin, then eluted by cleavage with TEV protease. The protein was further purified via affinity chromatography with a Mono Q HR 5/5 column (GE Healthcare).

The sequence encoding full-length human Pol β (Met 1-Glu335) was cloned into the SalI/NotI restriction sites of the pGEXT43(TEV) vector[41]. The vector was transformed into CodonPlus-RIL cells and expressed in LB supplemented with 100 μg/mL ampicillin and 35 μg/mL chloramphenicol. Cultures were grown at 37 °C to an $OD_{600nm}$ of 0.6–0.7, at which point the temperature was reduced to 18 °C for 40 min. Expression was induced by the addition of IPTG to a final concentration of 0.4 mM and allowed to proceed overnight. The cells were pelleted by centrifugation and lysed by sonication in 25 mM Tris pH 7.5, 500 mM NaCl, supplemented with 1 mM phenylmethylsulfonyl fluoride and complete (EDTA-free) protease inhibitor tablets (1 tablet per 40 mL of sonication buffer, Roche). The lysate was subsequently clarified by centrifugation, bound in-batch to glutathione sepharose 4B resin (Cytiva), washed extensively with sonication buffer, and subjected to on-resin TEV protease cleavage in 25 mM Tris pH 7.5, 75 mM NaCl overnight at 4 °C. Cleaved Polβ was concentrated and loaded onto a MonoS column in TEV cleavage buffer, followed by elution using a linear gradient to 25 mM Tris pH 7.5, 1 M NaCl. Fractions containing purified Pol β were pooled, dialyzed overnight at 4 °C to 25 mM Tris pH 8, 100 mM NaCl, concentrated to 18 mg/mL, and flash frozen in liquid nitrogen.

### Generation of a model of the NHEJ complex
A theoretical model of the NHEJ complex (based on PDB ID code: 7LSY)[12] was generated using AlphaFold-Multimer[42] to dock the BRCT domains of Pol μ (PDB ID code: 2HTF)[7] or Pol λ (PDB ID code: 2JW5)[8]. Models were visualized in PyMOL (Schrödinger, LLC, https://www.pymol.org).

### Detection of RNA fraction by alkali digestion
Harvested NHEJ products were treated with 300 mM KOH (or mock treated with 300 mM KCl) for 90 min at 60 °C. Samples were then neutralized by 4x dilution in 10 mM Tris, pH 8.5. supplemented with equimolar HCl (or H2O for mock-treated samples). Neutralized samples were then amplified by qPCR (see below), and RNA content was determined by the Ct difference between mock and alkali treatment.

$$\text{fraction RNA} = 1 - \left( 2^{(Ct_{mock} - Ct_{alkali})} \right)$$

This method was validated to be linearly responsive to RNA content by amplification of model NHEJ products comprised of exclusively DNA or containing a single embedded ribonucleotide, mixed in proportions ranging from 0-100% RNA-embedded products (Supplementary Fig. 3g).

### Crystallization of SSB 2nd strand intermediate with purified Pol λ catalytic domain
Catalytic domain constructs (Val235-Trp575) of human DNA Pol λ were purified as previously described[26]. Briefly, N-terminal Glutathion S-transferase fusion constructs were expressed in Rosetta2(DE3) cells after an overnight induction at 18 °C, and the catalytic domain purified from extracts by affinity chromatography using glutathione Sepharose 4B, followed by on-resin cleavage with Tobacco Etch Virus protease and size exclusion chromatography of the released catalytic domain. Purified proteins were kept in storage buffer containing 25 mM Tris, pH 8, 100 mM NaCl, 5% glycerol, 1 mM DTT, and concentrated to 12.74 mg/mL.

Oligonucleotides were used to generate the single-nucleotide gapped single-strand break (SSB) substrate representing the intermediate after Pol μ and Lig4 have facilitated ligation in the 1st strand of a noncomplementary DSB end by insertion of a ribonucleotide (Fig. 3c-d, top strand): template (5′-CGGCA**rG**TACTG-3′), upstream primer (5′-CAGTAC-3′), and 5′-phosphorylated downstream primer (5′-phosGCCG-3′). Oligonucleotides were mixed in an equimolar ratio in 100 mM Tris, pH 7.5, 40 mM MgCl2, and annealed in a thermal cycler, with denaturation at 94 °C, followed by a slow temperature gradient from 90 °C to 4 °C. The annealed DNA was then serially mixed in a 3:1 molar ratio with concentrated Pol λ catalytic domain (Val235-Trp575, 12.74 mg/mL)—followed by addition of the incoming nonhydrolyzable dUMPNPP nucleotide (0.91 mM final concentration). The complex was incubated on ice at 4 °C for 1 h, after each addition. Crystals of the ternary complex were grown at room temperature using the sitting drop vapor diffusion technique[43]. 400 nL of the protein/DNA complex was mixed with 200 nL of mother liquor (68 mM Na cacodylate, pH 6.5, 20.4% (w/v) PEG8000, 136 mM ammonium sulfate, 12% (v/v) glycerol). Drops containing crystals were opened, and 2× 1 μL cryoprotectant solution (82.9 mM Na cacodylate, pH 6.5, 24.9% (w/v) PEG8000, 166 mM ammonium sulfate, 14.6% (v/v) glycerol, 0.2 mM dUMPNPP, 10 mM MgCl2) was added to the drop. The crystals were looped directly from this drop and flash frozen in liquid nitrogen.

### Data collection and structural refinement
Crystals were placed into a stream of nitrogen gas cooled to −180 °C for data collection. Crystals of the Pol λ.5/1nt SSB 2nd strand intermediate complex were sent to the AMX beamline (17-ID-1)[44] at the National Synchrotron Light Source II (NSLSII) at Brookhaven National Laboratory for data collection. The crystals were placed into a stream of nitrogen gas cooled to -180 °C during the collection of diffraction data. The data were integrated and scaled using the autoPROC toolbox[45]. The phase problem was solved by molecular replacement in Phaser[46], using a previously published Pol λ SSB ternary complex structure (PDB ID code: 7M07)[26] as search model. The same $R_{free}$ test

set was used as for 7M07, to reduce model bias. The final structure was refined by iterative rounds of manual model building in COOT[47] and refinement in Phenix[48]. Translation/Libration/Screw (TLS) vibrational motion refinement was used[49]. Data collection and refinement statistics are listed in Supplementary Table 9. The Ramachandran statistics were generated by MolProbity[50]. All structural figures and superpositions were created using PyMOL[51]. Sugar puckering parameters were analyzed by Web 3DNA 2.0[52].

## Quantification and statistical analysis

Student t-tests (paired and unpaired) and one-way ANOVA with corrections for multiple comparisons (Tukey's and Dunnett's tests) were used where applicable. All statistical analysis was performed in GraphPad Prism 10, and the specific tests used for analysis are listed in the corresponding figure legends, along with values and representation of $n$ and $p$-values. Throughout the manuscript, $*p < 0.05$, $**p < 0.01$, $***p < 0.001$, and $****p < 0.0001$.

## Reporting summary

Further information on research design is available in the Nature Portfolio Reporting Summary linked to this article.

## Data availability

Raw data generated in this study for duplex sequencing experiments have been deposited in the NCBI Sequence Read Archive under accession code PRJNA1268600. Coordinates and structure factors have been deposited into the Protein Data Bank under PDB ID code 9NPU. Previously published data sets are publicly available as follows: 7M07[26], 7LSY[12], 2HTF[7], and 2JW5[8]. Source data are provided with this paper.

## Code availability

Analysis code is available at (https://github.com/aluthman/Ramsden-Lab.git) and at permanent repository Zenodo (https://doi.org/10.5281/zenodo.15528811)[53].

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

## Acknowledgements
We thank Ramsden lab members for helpful comments in reviewing this manuscript; Juno Krahn for assistance with modeling the NHEJ complex; Cameron Cordero for assistance in code development and sequence analysis; Dr. L. Blanco for providing mouse cell lines; Dr. E. Hendrickson for providing HCT116 cell lines; Dr. R. Wood for providing *Polq*⁻/⁻ mouse cells. National Cancer Institute grant R01 CA097096 (DAR). National Cancer Institute grant P01 CA247773 (DAR). National Institutes of Health grant 1ZIC ES102645 (LCP). National Institutes of Health grant Z01 ES065070 (TAK). This work was funded in part by the Division of Intramural Research of the National Institute of Environmental Health Sciences. This research used the AMX beamline (17-ID-1) of the National Synchrotron Light Source II, a U.S. Department of Energy (DOE) Office of Science User Facility operated for the DOE Office of Science by Brookhaven National Laboratory under Contract No. DE-SC0012704. The Center for BioMolecular Structure (CBMS) is primarily supported by the National Institutes of Health, National Institute of General Medical Sciences (NIGMS) through a Center Core P30 Grant (P30M133893), and by the DOE Office of Biological and Environmental Research (KP1605010).

## Author contributions
Conceptualization: A.J.L., K.K.C., D.A.R. Methodology: A.J.L., K.K.C., D.A.R. Investigation: A.J.L., K.K.C., A.M.K. Visualization: A.J.L., A.M.K. Funding acquisition: L.C.P., T.A.K., D.A.R. Project administration: D.A.R. Writing—original draft: A.J.L., D.A.R. Writing—review & editing: A.J.L., K.K.C., A.M.K., L.C.P., T.A.K., D.A.R.

## Competing interests
The authors declare no competing interests.
