## [Transparent Peer Review file · Nature Communications]

Nonhomologous end-joining uses distinct mechanisms to repair each strand of a double strand break

Corresponding Author: Dale Ramsden

Version 0:

Reviewer comments:

Reviewer #1

(Remarks to the Author)

This manuscript seeks to understand how symmetric and asymmetric double stranded DNA breaks are repaired via NHEJ. The authors use a novel mouse fibroblast assay in which various linear DNAs are electroporated into cells and assayed for repair via qPCR or next generation sequencing. The authors find that symmetrical DNAs are repaired in a parallel, symmetric manner whereas asymmetrical DNAs are repaired via an ordered, semi-conservative manner. Overall, this is an interesting and rigorous manuscript that should be of interest to the DNA repair field. Below I suggest a minimal number of additional assays that while not strictly required for publication would improve the clarity of the manuscript.

Major points:

- 1.) In Figure 2E, where one end has a CT mismatch, the repair pathway changes to favor ligation of GT and resection of the bottom strand which allows for re-polymerization of the bottom strand instead of incorporating the CT mismatch (as seen in the symmetric, dual GT mismatch substrate in 2B).
 - a.) How do the authors propose the NHEJ machinery is 'reading' the strands? It is confusing that direct ligation of the GT mismatch (which is arguably sterically larger) is favored. A structure of Lig4 in complex with CT vs GT would be ideal, but likely not possible in the context of this paper. Would the authors see the same effect by switching the position of the CT/GT mismatches? Is the sequence context around the CT or GT mismatch important? Adding these substrates would help in the interpretation of the findings.
 - b.) In figure 3, the authors do a nice job investigating the role of the NHEJ factors in repair to confirm that second strand ligation requires NHEJ. In Fig 2E it seems like second strand repair could also be independent of the NHEJ core machinery, especially given the 30 minute time scale for Model 2 repair. Is it possible that the mismatch repair machinery (Msh2-Msh6) is finding the mismatch afterward and repairing via MMR? Could the authors knockdown Msh2 to confirm the independence of MMR?
- 2.) In Figure 3D, the authors show that a Pol Mu dependent DNA substrate always undergoes repair by first adding a C to allow base pairing and ligation on one side.
 - a.) Can the authors test a less preferred nucleotide addition to stress the system and see if it still follows the same repair path?
 - b.) The authors should also use a pre-added dCTP instead of rCTP on the end to see if repair timing or outcome is changed. Their model and structure suggest that it will not, but the lack of rNTP may make a difference in affinity for replicative polymerases.
- 3.) In their proposed model, the authors show "DNA rotation" as a means by which ligation and processing are allowed within the short range complex. Isn't a simpler model that Lig4 / polymerases are able to reorient themselves within the complex given their flexible linkers? The authors should present both possibilities in their discussion.

Minor points:

- 1.) It would be nice to see the 30 minute data for Figure 2C, as done with 2E.
- 3.) Figure 3E is difficult to interpret – what do the different sides of the dotted line represent? What is the time scale looked at in the figure?

(Remarks on code availability)

Reviewer #2

(Remarks to the Author)

In this study, the authors transfect linear DNA molecules into human cell lines or mouse embryonic fibroblasts (MEFs) to examine the ligation of the top and bottom strands of the two DNA ends of the linear DNA molecules. They have mismatches in the top and bottom strands that would be joined from the same linear substrate (intramolecular joining). The mismatches allow the authors to distinguish the top and bottom strands. The authors harvest the nucleic acid at short time points (e.g., 2 minutes, 30 minutes) after transfection and analyze the products by PCR (vast excess of diverse primers). The authors infer that the DNA end state dictates the steps involved in the joining of the top and bottom strands. For ends requiring only minimal or no trimming by a nuclease, their model 1 is mostly used. For ends that require nuclease action or polymerase addition, then the possibilities broaden to include model 2, even to the point of being predominant.

This is a novel and interesting study. I have a few minor questions below. For the benefit of the authors and readers, I insist that they add at least one paragraph (likely more) discussing the limitations and qualifications of this approach. This will only help the authors if future studies find anything that is different. This will help readers appreciate the pros and cons of various ways by which one can approach these challenging topics.

Minor Questions.

1. Can the authors comment on whether intermolecular [rather than intramolecular circularization (head to tail)] joining would affect the analysis? I suspect not a problem, but I want to make sure.
2. Electroporation creates many electrophoresis on two sides of each cell. Obviously, for the linear DNA to enter, the cells, the pores are quite large (at least 20 to 100 fold larger than ATP). I am concerned about ATP (and other nucleotide) depletion, which may affect the re-charging of ligase 4. The authors may wish to include ATP in the electroporation buffer to ensure that this is not affecting any of their findings. I do not insist on this for this study, but the authors may find it to be a useful reassurance.
3. Do the authors think that two ligase 4 molecules are present at each double-strand break (DSB)? If two ligase 4 are present in their charged state, re-charging (as shown in the video) may not be necessary. Note that re-charging of the ligase 4 involves other proteins that would need to gain access to the DSB junction.

Limitations.

For the limitations paragraphs, the authors should list the following.

- a. Some NHEJ proteins (probably most) are located in the nucleus. It is likely that many or most of the DNA substrates do not arrive at the nucleus in 2 minutes. If some or most of the joining events are occurring outside of the nucleus, then differential involvement of some NHEJ proteins (e.g., NHEJ nucleases needed to trim the ends, such as the complex of Artemis:DNA-PKcs) may not be present. This may skew the results.
- b. Chromatin is not present, or if present, such as abundant H1, then this may alter the protein access to the substrate.

References.

PMID 17717001 and 27703001 - using purified NHEJ proteins, these examine the ligation of one strand rather than both for incompatible ends or ends with blockage.

(Remarks on code availability)

Reviewer #3

(Remarks to the Author)

In this manuscript from the Ramsden laboratory, numerous clever strategies are utilized to determine the mechanistic steps that lead to ligation of the two different strands in DNA end joining substrates with structurally distinct termini (mimicking chromosomal DNA double-strand breaks). The authors posit two methods for the ligation step in NHEJ-mediated joining -- either symmetrical joining of the two strands, or asymmetrical joining of one strand, leaving a single-stranded break that theoretically could be joined by SSB, independent of NHEJ. In fact, the unique characteristic of the canonical NHEJ ligase (ligase 4, L4) as the only single turnover ligase in higher organisms might predict that each strand of a DNA double-stranded break would need to be addressed individually, and thus potentially by distinct mechanisms. (The authors might discuss this connection.)

In any case, their results are quite striking. This study discovers that both symmetrical and asymmetrical joining of the two strands occurs during NHEJ in living cells. Moreover, the DNA end structure dictates the mechanism of joining. They

conclude that strand breaks that can be positioned so that ligation is possible are symmetrically and almost simultaneously ligated. In the case of asymmetric NHEJ-mediated joining, the re-joined strand serves as a template for repair of the second strand. To my knowledge, this is the first evidence that the NHEJ associated polymerases utilize the repaired strand as a template. Additional experiments define pol as the polymerase that inserts an RNA nucleotide to facilitate end pairing, pol as the polymerase that utilizes the re-joined first strand as a repair template, and RNaseH1 as the enzyme that removes RNA bases from re-joined ends. In sum, these results provide important insight into how end-processing in the context of the NHEJ short-range complex proceeds.

Suggestions:

1. NHEJ-mediated joining of incompatible DNA ends always results in somewhat diverse joining events. Although characterized as an error-prone pathway, emerging studies (including previous studies from these authors) demonstrate that NHEJ highly prefers to generate the least mutagenic joint possible. It would be helpful for the reader to have information regarding the relative frequencies for preferred joints presented in each figure. For instance, it is easy to miss the information in results text that 93% of the joints with the first substrate are direct joining of the mispaired ends; and it follows that the analysis of strand joining in Fig 1A-C only addresses joining of this particular joint. With the other substrates, it is less clear what fraction of the joined substrates the joint examined represents. I could find no information as to the percentage of the joint described in Fig1D-E. For the non-complementary (GAG) substrates this is provided in supplemental data. The authors should present this information in each figure or at least clearly state that only a defined type of joint is analyzed from each substrate. Without clarifying this issue, a more casual reader might be confused as to how this nice strategy works.

2. In figure 2 (A-C), with the symmetrical substrate, why is there such a big difference in the joining of only the top strand (20%) versus the bottom strand (7.8%)? In any case, this difference also argues that joining of each strand is handled separately, but in a situation where end chemistry allows positioning of the ends for ligation, no fill-in processing occurs.

3. In figure 2 (D-E), the single nucleotide difference in this substrate, introducing a T/C mismatch on one side as opposed to G/T mismatches on both sides results in a striking difference in sequences of rejoined ends. This is evidenced by the rate of single joining of the T/C mismatch end (after 30 minutes) remarkably reduced compared to the G/C mismatch. Is this because the mismatch is pyrimidine/pyrimidine mismatch as opposed to the G/T mismatch? A comment regarding the decreased joining of this side of the break should be included. And also, as noted above, what percentage of the recovered joints do these directly ligated ends now represent? Still 93%?

The observation that 44% of the recovered breaks join the second strand by using the first repaired strand as a template is compelling; clearly the ligations of the two strands at a single break are temporally distinct.

4. There should be a new heading in the results text for presentation of figure 3.

5. At the end of the introduction the authors state: Here we investigate the relationship between repair of each of the two strand breaks of a DSB by NHEJ, as well as the role of the end-bridging complex in organizing repair of each strand. Only the first part of this sentence is really accurate. There are no experiments that address end-bridging. Elsewhere in the manuscript, different end-bridging complexes are discussed in the context of the elegant end-joining experiments presented in this manuscript. Although there are numerous long-range complexes where ends are held too far apart to be ligated, only one short-range complex (termed end-bridging complex in this manuscript) has been reported. The elegant discussion of the structural considerations of first and second strand joining within this complex is insightful, though it would be useful if the authors referred to this complex as a short-range complex, the terminology that seems to be most widely accepted in the field today.

Minor comments:

1. A cartoon of the exact sequence of +C product should be in figure 3. It is confusing that joining rates in 3B are relative to wild type, whereas joining rates in 3E are apparently absolute values.

2. There are several sentences that seem awkward, although potentially still grammatically correct. The authors might consider rewording the following:

The unique ability to sustained synthesis activity in this context relies....

We track the repair of each strand in living cells over time, using methods that allow identification of both which strand is being repaired and the sequence of that strand's repair product (Fig. 1B) with single molecule resolution.

Structural studies indicate these complexes elegantly help address

(Remarks on code availability)

Version 1:

Reviewer comments:

Reviewer #1

(Remarks to the Author)

The authors have answered all of my concerns, which were minor to begin with. Congrats to the authors on a wonderful story!

(Remarks on code availability)

Reviewer #2

(Remarks to the Author)
Accept.

(Remarks on code availability)

Reviewer #3

(Remarks to the Author)
Great job revising this nice manuscript.

(Remarks on code availability)

We would like to thank the referees for their careful review and helpful comments. Though all referees suggested additional experiments were “not strictly necessary” we very much appreciate their concerns. Accordingly, we have added new experiments reported in Figs. S2B-E and Fig. S3F, and include new data (Tables S3, S7, and S8) that more fully describe previous experiments. We have additionally made significant changes to the text, with changes tracked in this response using line references to the edited document, as well as in the edited document using the “track changes” feature in MS Word (as requested by the journal). Please see below the verbatim comments from each referee in red, with our point-by-point responses directly following each concern that the referees raised.

Reviewer #1 (Remarks to the Author):

This manuscript seeks to understand how symmetric and asymmetric double stranded DNA breaks are repaired via NHEJ. The authors use a novel mouse fibroblast assay in which various linear DNAs are electroporated into cells and assayed for repair via qPCR or next generation sequencing. The authors find that symmetrical DNAs are repaired in a parallel, symmetric manner whereas asymmetrical DNAs are repaired via an ordered, semi-conservative manner. Overall, this is an interesting and rigorous manuscript that should be of interest to the DNA repair field. Below I suggest a minimal number of additional assays that while not strictly required for publication would improve the clarity of the manuscript.

Major points:

1.) In Figure 2E, where one end has a CT mismatch, the repair pathway changes to favor ligation of GT and resection of the bottom strand which allows for re-polymerization of the bottom strand instead of incorporating the CT mismatch (as seen in the symmetric, dual GT mismatch substrate in 2B).

a.) How do the authors propose the NHEJ machinery is ‘reading’ the strands? It is confusing that direct ligation of the GT mismatch (which is arguably sterically larger) is favored. A structure of Lig4 in complex with CT vs GT would be ideal, but likely not possible in the context of this paper. Would the authors see the same effect by switching the position of the CT/GT mismatches? Is the sequence context around the CT or GT mismatch important? Adding these substrates would help in the interpretation of the findings.

As also noted in a response to another referee, we’d previously reported LIG4 favors G:T mispairs – especially when the G:T mispair is on the 5’ side of the strand break¹, as employed here. The 5’G:T mispair is one of the few examples of ligase infidelity that is frequently observed across different ligases², possibly because hydrogen bonds from a “wobble” base confirmation is possible. Regarding the “..sterically larger...” point – we presume this is why most purine:purine mispairs are the least tolerated (we almost never observe direct ligation of these mispairs). We chose C:T because we expected it to be intermediate in its ability to be directly ligated: less well tolerated than G:T. but not necessarily “rejected”, like most purine-purine mispairs.

We agree additional substrates would be helpful in validating ligation proficiency. We’ve therefore added data for two new substrates, both the switching experiment requested, as well as a substrate with 5’ terminal C:T mispairs in both strands (new figs. S2B-E). Our results confirm C:T mispairs are less efficiently joined by direct ligation regardless of context (lines 135-137).

As a point of interest (and as pointed out in response to another referee's point), the flanking sequence context may indeed also play a role, though it's more subtle. There are two examples of where early repair of a specific strand is favored despite symmetric overhang sequence. The top strand is more often repaired before the bottom strand repair in one case (Fig. 2B-D), while the bottom strand is more often repaired before the top strand in another (Fig. S3A).

b.) In figure 3, the authors do a nice job investigating the role of the NHEJ factors in repair to confirm that second strand ligation requires NHEJ. In Fig 2E it seems like second strand repair could also be independent of the NHEJ core machinery, especially given the 30 minute time scale for Model 2 repair. Is it possible that the mismatch repair machinery (Msh2-Msh6) is finding the mismatch afterward and repairing via MMR? Could the authors knockdown Msh2 to confirm the independence of MMR?

We note i) the mispaired region we use to discriminate strands is stable for 30 minutes (Fig. S1F; this is why experiments are limited to this duration) and ii) ligation of G:T mispairs in both strands is also stable (Fig. 2C compared to 2D), iii) a role for mismatch repair would argue there should be an early burst in ligation of the C:T pair before the mismatch is eventually corrected (we instead see an excess of unrepaired bottom strand at early time points). Finally, iv) we recently repeated this experiment in mismatch repair deficient HCT116 cells (MLH1 deficient), and confirmed Model 2 repair remains the preferred path for the bottom strand. We should emphasize we view this latter experiment is at-best suggestive. HCT116 is human, a different species than that used in the cell lines used in this manuscript, and we do not have a wild type complemented control cell line to compare it to.

We nevertheless appreciate this point, and agree there are multiple NHEJ-independent mechanisms that can (and at least eventually, will, if NHEJ does not) generate a repaired second strand now that the first has already been repaired. In accord with this idea, we emphasize how our data presented in Fig. 3 show that repair of the 2nd strand by model 2 is "leakier" than 1st strand repair, in terms of NHEJ dependence, (compare impact of Pol mutants on 1st strand, Fig. 3B, to 2nd strand, Fig. 3E). However, we suspect the most likely source of the NHEJ independent background is single strand break repair, and anticipate rigorously addressing this possibility in future work.

2.) In Figure 3D, the authors show that a Pol Mu dependent DNA substrate always undergoes repair by first adding a C to allow base pairing and ligation on one side.

a.) Can the authors test a less preferred nucleotide addition to stress the system and see if it still follows the same repair path?

We provide additional data indicating that only the +C product is biologically significant (lines 157-158). Additions of other nucleotides were not detected. Given detection limits, this indicates their frequencies are at least 1000-fold less common than +C.

We also note here we had previously employed an in vitro reconstituted system to follow Pol μ activity during NHEJ, and used chain terminators to ensure we could track addition by Pol μ , whether or not the addition is a good substrate for ligation. Pol μ adds complementary C opposite template G at least 10 fold more often than the other 3 possibilities³, and additionally favors the first single stranded template position immediately upstream of a double stranded terminus for structural reasons^{4,5}.

In sum, Pol μ rarely adds nucleotides other than a single C for the substrate in question.

Moreover, if added, these other nucleotides do not contribute significantly to cellular repair.

b.) The authors should also use a pre-added dCTP instead of rCTP on the end to see if repair timing or outcome is changed. Their model and structure suggest that it will not, but the lack of rNTP may make a difference in affinity for replicative polymerases.

Pre-added dC does impact timing of repair, as well as outcome. Joining is inefficient and delayed with pre-added dC. This is partly because ribonucleotide termini in the contexts where Pol μ and TdT are active (partially mispaired ends) are a preferred substrate for the ligation step, as indicated in a previous report (in citation⁶, specifically Fig 3, Table 1, and Fig S5 of this publication). This ability of a riboN 3' terminus, but not a deoxyN 3' terminus, to stimulate ligation is mentioned in results text (lines 181-182): "...addition of RNA instead of DNA is essential for accurate and efficient joining by NHEJ of this class of substrates⁶." We have now added a figure with the requested experiment (Fig. S3F), and include a citation of this new figure as additional evidence for this.

Interestingly, this context-dependent favoring of ribonucleotide 3' termini by LIG4 has precedent in some of the prokaryotic ligases that perform NHEJ^{7,8}, but is not observed in "non-NHEJ" prokaryotic ligases (e.g. T4 DNA ligase⁶).

3.) In their proposed model, the authors show "DNA rotation" as a means by which ligation and processing are allowed within the short range complex. Isn't a simpler model that Lig4 / polymerases are able to reorient themselves within the complex given their flexible linkers? The authors should present both possibilities in their discussion.

We agree this section needs to be expanded, and now elaborate more on why a DNA motion model is favored (lines 276-284). We also emphasize here that this aspect of the model is based on comments made in both of the relevant cryo-EM structure publications^{9,10}. In our initial submission we had only cited the most recent of these publications. We have now added the older citation and apologize for this omission.

In some more detail here: all existing cryo-EM structures that are catalytically permissive (short range complexes) position the active strand break in roughly the same place, on the "top" of the omega/short range complex, opposite dimers of XLF and XRCC4 and the "bottom" strand break. In this location there's sufficient volume for relevant active sites (LIG4 NTD subdomain or Pol μ active site) to engage the top strand break termini. However, there isn't sufficient volume on the opposite side of the helix (the bottom strand) to be catalytically engaged, especially given the "kinks" in helix path associated with catalytic activity (35° for LIG4, 80° for Pol μ).

Moreover, both of the cryo-EM studies identify protein-protein interactions between the 1st subdomain of LIG4 (the DBD) and Ku, XLF, and (when present) Pol μ . These interactions help fix the location of the DBD underneath the bottom strand. The short range complex space-filling representations in Figure 6 of Liu et al¹⁰ help illustrate the points made above.

The DNA helix thus has to move relative to the rest of the complex if the 2nd strand break is to be productively engaged. The helical phase of the 2nd strand break, relative to the 1st strand break, determines whether the helix needs to rotate (or change helical twist), translocate, or both, such that the bottom strand break is relocated to the top and near-center. Given that most biological substrates for NHEJ have <5 nucleotide staggers between the two strand break termini, we argue the major required DNA motion will be rotation.

Minor points:

1.) It would be nice to see the 30 minute data for Figure 2C, as done with 2E.

We agree, and had previously included this only as a supplement (old Fig. S2B) as it's consistent with bulk qPCR data already present as display Fig. 2B, didn't impact our conclusions, and its

inclusion in the display figure made initial attempts at a good figure layout challenging. We have now included this data in the display figure (new Fig. 2D), pending journal approval of the overall figure layout.

3.) Figure 3E is difficult to interpret – what do the different sides of the dotted line represent? What is the time scale looked at in the figure?

For Figs. 3B and 3E we've revised the legend text to more clearly state that the dotted line separates independent experiments, and that this experiment addresses products generated after 5 minutes in cells (lines 793-794 and 815-820).

Reviewer #2 (Remarks to the Author):

In this study, the authors transfect linear DNA molecules into human cell lines or mouse embryonic fibroblasts (MEFs) to examine the ligation of the top and bottom strands of the two DNA ends of the linear DNA molecules. They have mismatches in the top and bottom strands that would be joined from the same linear substrate (intramolecular joining). The mismatches allow the authors to distinguish the top and bottom strands. The authors harvest the nucleic acid at short time points (e.g., 2 minutes, 30 minutes) after transfection and analyze the products by PCR (vast excess of diverse primers). The authors infer that the DNA end state dictates the steps involved in the joining of the top and bottom strands. For ends requiring only minimal or no trimming by a nuclease, their model 1 is mostly used. For ends that require nuclease action or polymerase addition, then the possibilities broaden to include model 2, even to the point of being predominant.

This is a novel and interesting study. I have a few minor questions below. For the benefit of the authors and readers, I insist that they add at least one paragraph (likely more) discussing the limitations and qualifications of this approach. This will only help the authors if future studies find anything that is different. This will help readers appreciate the pros and cons of various ways by which one can approach these challenging topics.

Minor Questions.

1. Can the authors comment on whether intermolecular [rather than intramolecular circularization (head to tail)] joining would affect the analysis? I suspect not a problem, but I want to make sure.

We agree, we don't see how it could impact our interpretations – the aligned intermediates and product structures are the same regardless, and our substrates are sufficiently long that both inter and intra molecular ligations are at least possible (very short dsDNA cannot circularize, at least *in vitro*). We can note to the referee that the majority of products we recover are sensitive to exonucleases. While we don't view this evidence as definitive, it's at least suggestive that products are primarily linear (not circles). This is consistent with many (I'd suggest most) *in vitro* models.

2. Electroporation creates many electrophoresis on two sides of each cell. Obviously, for the linear DNA to enter, the cells, the pores are quite large (at least 20 to 100 fold larger than ATP). I am concerned about ATP (and other nucleotide) depletion, which may affect the re-charging of ligase 4. The authors may wish to include ATP in the electroporation buffer to ensure that this is not affecting any of their findings. I do not insist on this for this study, but the authors may find it to be a useful reassurance.

We can confirm that addition of 1 mM ATP in the electroporation buffer does not significantly impact our results or interpretations (see figure below).

Figure for review. The substrate in Fig. 2A of the manuscript was introduced into T-antigen transformed mouse embryo fibroblasts (MEFs) in electroporation buffer with 1 mM ATP (+) or without added ATP (-) and DNA was harvested after 30 minutes. Greater repair is represented increasing upwards from the y axis origin for the top strand or increasing downwards from the origin for the bottom strand. The amounts of directly ligated strands are plotted for the top strand in green and the bottom strand in orange. Products using an opposite strand as template (model 2) are plotted for the top strand in yellow and the bottom strand in blue. All repair amounts are scaled relative to the most abundant strand (top strand, all products) at 30 minutes as determined by digital droplet PCR.

3. Do the authors think that two ligase 4 molecules are present at each double-strand break (DSB)? If two ligase 4 are present in their charged state, re-charging (as shown in the video) may not be necessary. Note that re-charging of the ligase 4 involves other proteins that would need to gain access to the DSB junction.

We agree this is an interesting question, and discuss this in lines 287-290. In some more detail here: the available cryo-EM structures have two molecules of LIG4 in the short range complex^{9, 10}. However, a single molecule FRET experiment argues there is only one molecule of LIG4 present immediately prior to short range complex formation and presumed catalysis¹¹. This latter result also bears on the referee's comment, as the apparently dynamic nature of LIG4 association suggests a "spent" LIG4 molecule would more easily be exchanged for a fresh one, rather than re-adenylating the first.

Limitations. For the limitations paragraphs, the authors should list the following.

a. Some NHEJ proteins (probably most) are located in the nucleus. It is likely that many or most of the DNA substrates do not arrive at the nucleus in 2 minutes. If some or most of the joining events are occurring outside of the nucleus, then differential involvement of some NHEJ proteins (e.g., NHEJ nucleases needed to trim the ends, such as the complex of Artemis:DNA-PKcs) may not be present. This may skew the results.

b. Chromatin is not present, or if present, such as abundant H1, then this may alter the protein access to the substrate.

We also agree it is important to consider possible limitations of experimental models, and have expanded the initial summary of experimental rationale to make note of these limitations (lines 332-338).

References.

PMID 17717001 and 27703001 - using purified NHEJ proteins, these examine the ligation of one strand rather than both for incompatible ends or ends with blockage.

We appreciate the importance of the noted citations. Both PMID 17717001 and 27703001 as well as many other foundational studies addressing repair of diverse end structures are referenced in the review that we cite¹².

This manuscript argues that in cells, both strands must be repaired to maintain genome stability (lines 224-227), and we show that cellular NHEJ largely addresses this problem for the substrates employed here. We agree further exploration of this question in cells using additional substrates, including blocks (like the ones used in the citations above, and others, e.g. radiomimetic damage), is an important next step, and have added this note in manuscript text (line 334-335).

Reviewer #3 (Remarks to the Author):

In this manuscript from the Ramsden laboratory, numerous clever strategies are utilized to determine the mechanistic steps that lead to ligation of the two different strands in DNA end joining substrates with structurally distinct termini (mimicking chromosomal DNA double-strand breaks). The authors posit two methods for the ligation step in NHEJ-mediated joining -- either symmetrical joining of the two strands, or asymmetrical joining of one strand, leaving a single-stranded break that theoretically could be joined by SSBR, independent of NHEJ. In fact, the unique characteristic of the canonical NHEJ ligase (ligase 4, L4) as the only single turnover ligase in higher organisms might predict that each strand of a DNA double-stranded break would need to be addressed individually, and thus potentially by distinct mechanisms. (The authors might discuss this connection.)

In any case, their results are quite striking. This study discovers that both symmetrical and asymmetrical joining of the two strands occurs during NHEJ in living cells. Moreover, the DNA end structure dictates the mechanism of joining. They conclude that strand breaks that can be positioned so that ligation is possible are symmetrically and almost simultaneously ligated. In the case of asymmetric NHEJ-mediated joining, the re-joined strand serves as a template for repair of the second strand. To my knowledge, this is the first evidence that the NHEJ associated polymerases utilize the repaired strand as a template. Additional experiments define polm as the polymerase that inserts an RNA nucleotide to facilitate end pairing, poll as the polymerase that utilizes the re-joined first strand as a repair template, and RNaseH1 as the enzyme that removes RNA bases from re-joined ends. In sum, these results provide important insight into how end-processing in the context of the NHEJ short-range complex proceeds.

Suggestions:

1. NHEJ-mediated joining of incompatible DNA ends always results in somewhat diverse joining events. Although characterized as an error-prone pathway, emerging studies (including previous studies from these authors) demonstrate that NHEJ highly prefers to generate the least mutagenic joint possible. It would be helpful for the reader to have information regarding the relative frequencies for preferred joints presented in each figure. For instance, it is easy to miss the information in results text that 93% of the joints with the first substrate are direct joining of the mispaired ends; and it follows that the analysis of strand joining in Fig 2A-C only addresses joining of this particular joint. With the other substrates, it is less clear what fraction of the joined substrates the joint examined represents. I could find no information as to the percentage of the joint described in Fig2D-E. For the non-complementary (GAG) substrates this is provided in supplemental data. The authors should present this information in each figure or at least clearly

state that only a defined type of joint is analyzed from each substrate. Without clarifying this issue, a more casual reader might be confused as to how this nice strategy works.

We agree this is important information and apologize for its omission. We have added new Tables (supplemental Tables S3, S8, and S9) to include this information, as well as results text referring to it (lines 92, 131, and 151). We additionally now express this data in a slightly different manner, as it is better reconciled with our experimental approach, and also specifically addresses this reviewer's comment 3 below. We express the % of a product sequence as a fraction of all NHEJ products **for a specific strand** (the "top", or FAM+ strand), and the tables include this information for the substrates used in Figure 2B-D (Table S3), Fig 2D-E (Table S8), as well as Figure 3A-D (the GAG data; Table S9). When analyzed this way, we observe 96% direct ligation for the substrate used in Fig 2A-D (not 93%).

We also include the 5 most frequent sequences recovered for the top strands for these three substrates in this table. We emphasize that the detection limit in these sequencing experiments for a given junction sequence is $\sim 10^{-4}$, as we intentionally limited the number of input template molecules, to ensure we could definitively interpret the duplex sequencing results (the UMI depth of $\sim 10^6$ has to be ~ 100 fold greater than the number of template molecules in a sample, to have confidence in single duplex molecule origin).

2. In figure 2 (A-C), with the symmetrical substrate, why is there such a big difference in the joining of only the top strand (20%) versus the bottom strand (7.8%)? In any case, this difference also argues that joining of each strand is handled separately, but in a situation where end chemistry allows positioning of the ends for ligation, no fill-in processing occurs.

We appreciate this reviewer's attention to detail. As noted, ligation of the bottom strand for the relevant substrate is reproducibly delayed, despite the symmetry in overhang sequence (Fig. 2B, 2C).

We would like to also point out that there is a similar phenomenon evident in joining for the symmetric overhang-containing substrate employed in Figure 3A-D. In this case we reproducibly see favored early repair of the bottom strand (cells accumulate twice as much +C addition on the bottom strand, relative to +C addition on the top strand; Fig. S3A). Importantly, for both of the 1st strand repair products (top strand 1st, as well as bottom strand 1st), the second strand is repaired through a model 2 path (Fig. S3E).

In short, for both substrates it doesn't matter which strand is repaired first – the second strand is repaired by a common mechanism (model 1 for Fig. 2A-D; Model 2 for Fig. 3). Thus, as the referee notes, these observations of asymmetry in early activity are surprising, but do not impact our interpretations.

It's worth noting that while the overhangs are symmetric, the flanking double stranded DNA isn't, and this may be sufficient to drive the modest asymmetries we see in which strand is the 1st to be catalytically engaged.

3. In figure 2 (D-E), the single nucleotide difference in this substrate, introducing a T/C mismatch on one side as opposed to G/T mismatches on both sides results in a striking difference in sequences of rejoined ends. This is evidenced by the rate of single joining of the T/C mismatch end (after 30 minutes) remarkably reduced compared to the G/T mismatch. Is this because the mismatch is pyrimidine/pyrimidine mismatch as opposed to the G/T mismatch? A comment regarding the decreased joining of this side of the break should be included. And also, as noted above, what percentage of the recovered joints do these directly ligated ends now

represent? Still 93%? The observation that 44% of the recovered breaks join the second strand by using the first repaired strand as a template is compelling; clearly the ligations of the two strands at a single break are temporally distinct.

As also noted in a response to another referee, we'd previously reported LIG4 favors G:T mispairs over many other mispairs – especially when the G:T mispair is on the 5' side of the strand break¹, as assessed here. Efficient ligation of G:T mispairs is also seen in other ligases², and it's been suggested this is because it allows for wobble base pairing with minimal helical distortion.

Since this was also commented on by Referee 1, we now provide additional experiments confirming direct ligation of C:T terminal mispairs is less efficient, relative to G:T mispairs, regardless of context (whether the C:T mispair is in the top strand, the bottom strand, or both; new Fig. S2E, line 137)

Regarding the request for us to comment on decreased joining of this strand – we've elaborated on our previous version of the relevant statements (lines 134-137)

As requested above, we now report the frequency of the 5 most frequent products observed for the top strand of the substrate employed in Fig. 2E-F. Regarding the fraction of top strand joints that involve direct ligation of a G:T mispair for the substrate in Fig. 2E-F, it's also 96% (new Table S8).

4. There should be a new heading in the results text for presentation of figure 3..

The current header titles, as well as our division of the manuscript with these headers, was chosen as we considered them the most consistent with our experimental question (whether model 1 or model 2 paths are used for repairing both strands, as shown in the experiments described in the paragraphs associated with header 1, vs. the paragraphs associated with header 2, respectively). A third header would orphan Fig. 3 experiments from its appropriate header (header 2) and leave Fig. 2E-F experiments described in a single paragraph header-separated section.

We prefer this organization over other possibilities (section headers according to substrate end structure, perhaps) but can change this if the referee remains confident a third header is important.

5. At the end of the introduction the authors state: Here we investigate the relationship between repair of each of the two strand breaks of a DSB by NHEJ, as well as the role of the end-bridging complex in organizing repair of each strand. Only the first part of this sentence is really accurate.

We'd intended this part of the sentence to refer to our experiments using the two PolX mutants that don't efficiently interact with NHEJ core factors (the end-bridging complex). We interpret these experiments as permitting the dissection of core factor (and thus end-bridging complex) contributions to 1st vs. 2nd strand repair, as the different polymerases are active on different strands. We have rephrased the relevant statement to more accurately reflect this experimental approach (line 62-64).

There are no experiments that address end-bridging. Elsewhere in the manuscript, different end-bridging complexes are discussed in the context of the elegant end-joining experiments presented in this manuscript. Although there are numerous long-range complexes where ends are held too far apart to be ligated, only one short-range complex (termed end-bridging complex in this manuscript) has been reported. The elegant discussion of the structural considerations of first and second strand joining within this complex is insightful, though it would be useful if the authors

referred to this complex as a short-range complex, the terminology that seems to be most widely accepted in the field today.

We note they are several additional short range complex structures in a recent publication¹⁰, and this publication also adds another term for them (the ω complex, after its shape).

As the referee notes, we focus in the discussion on a model that features a primary role for short-range complexes in the events we study here. Our revised text now more consistently refers to the short-range complex, in keeping with the widely accepted terminology.

However, we note we also consider a possible role for long-range complexes when switching strands (lines 311-316). We thus thought it appropriate to use the more generic (and, in our opinion, intuitive) term “end-bridging complexes” whenever the text didn’t require additional specificity. We don’t feel strongly about this distinction, however, and can adjust text if the reviewer remains concerned about wide readability.

Minor comments:

1. A cartoon of the exact sequence of +C product should be in figure 3.

The 3A panel previously had a cartoon describing this product - we assume the primary concern here is that the axis label references to “+C” product in subsequent panels are too cryptic. We’ve therefore added a label to the 3A panel identifying the relevant product in the cartoon, as well as legend text for subsequent Fig. 3 panels to improve clarity. If the concern is that there isn’t enough sequence context in the 3A panel cartoon, we’ve included a little more context in the new supplementary table S9.

It is confusing that joining rates in 3B are relative to wild type, whereas joining rates in 3E are apparently absolute values.

The axis label and legend text previously used for Fig. 3E poorly described what we measured. We’ve re-labeled the axis and have edited the legend text (lines 816-820) to state explicitly that Fig. 3E is a ratio of two absolute (ddPCR generated) numbers.

2. There are several sentences that seem awkward, although potentially still grammatically correct. The authors might consider rewording the following:

The unique ability to sustained synthesis activity in this context relies....

We track the repair of each strand in living cells over time, using methods that allow identification of both which strand is being repaired and the sequence of that strand's repair product (Fig. 1B) with single molecule resolution.

Structural studies indicate these complexes elegantly help address

The noted sentences have been shortened and/or split, hopefully making them at least a little less awkward.

BIBLIOGRAPHY AND REFERENCES CITED

1. Waters CA, *et al.* The fidelity of the ligation step determines how ends are resolved during nonhomologous end joining. *Nat Commun* **5**, 4286 (2014).
2. Lohman GJ, *et al.* A high-throughput assay for the comprehensive profiling of DNA ligase fidelity. *Nucleic Acids Res* **44**, e14 (2016).

3. Davis BJ, Havener JM, Ramsden DA. End-bridging is required for pol mu to efficiently promote repair of noncomplementary ends by nonhomologous end joining. *Nucleic Acids Res* **36**, 3085-3094 (2008).
4. Moon AF, Gosavi RA, Kunkel TA, Pedersen LC, Bebenek K. Creative template-dependent synthesis by human polymerase mu. *Proc Natl Acad Sci U S A* **112**, E4530-4536 (2015).
5. Pryor JM, *et al.* Essential role for polymerase specialization in cellular nonhomologous end joining. *Proc Natl Acad Sci U S A* **112**, E4537-4545 (2015).
6. Pryor JM, *et al.* Ribonucleotide incorporation enables repair of chromosome breaks by nonhomologous end joining. *Science* **361**, 1126-1129 (2018).
7. Unciuleac MC, Goldgur Y, Shuman S. Structures of ATP-bound DNA ligase D in a closed domain conformation reveal a network of amino acid and metal contacts to the ATP phosphates. *J Biol Chem* **294**, 5094-5104 (2019).
8. Zhu H, Shuman S. Characterization of *Agrobacterium tumefaciens* DNA ligases C and D. *Nucleic Acids Res* **35**, 3631-3645 (2007).
9. Chen S, *et al.* Structural basis of long-range to short-range synaptic transition in NHEJ. *Nature* **593**, 294-298 (2021).
10. Liu L, *et al.* Dynamic assemblies and coordinated reactions of non-homologous end joining. *Nature* **643**, 847-854 (2025).
11. Stinson BM, Carney SM, Walter JC, Loparo JJ. Structural role for DNA Ligase IV in promoting the fidelity of non-homologous end joining. *Nat Commun* **15**, 1250 (2024).
12. Zhao B, Rothenberg E, Ramsden DA, Lieber MR. The molecular basis and disease relevance of non-homologous DNA end joining. *Nat Rev Mol Cell Biol* **21**, 765-781 (2020).